biomechanics/biomimetics/robotics

neuromuscular models, bipedal locomotion, biarticular muscles, reflex-based control, template and anchor

**Author for correspondence:**
Maziar A. Sharbafi
e-mail: sharbafi@sport.tu-darmstadt.de

# From template to anchors: transfer of virtual pendulum posture control balance template to adaptive neuromuscular gait model increases walking stability

Ayoob Davoodi[1], Omid Mohseni[1], Andre Seyfarth[2] and Maziar A. Sharbafi[1,2]

[1]School of ECE, Control and Intelligent Processing Center of Excellence (CIPCE), College of Engineering, University of Tehran, Tehran, Iran
[2]Lauflabor Locomotion Lab, Centre for Cognitive Science, TU Darmstadt, Germany

MAS, 0000-0001-5727-7527

Biomechanical models with different levels of complexity are of advantage to understand the underlying principles of legged locomotion. Following a minimalistic approach of gradually increasing model complexity based on *Template & Anchor* concept, in this paper, a spring-loaded inverted pendulum-based walking model is extended by a rigid trunk, hip muscles and reflex control, called nmF (neuromuscular force modulated compliant hip) model. Our control strategy includes leg force feedback to activate hip muscles (originated from the FMCH approach), and a discrete linear quadratic regulator for adapting muscle reflexes. The nmF model demonstrates human-like walking kinematic and dynamic features such as the virtual pendulum (VP) concept, inherited from the FMCH model. Moreover, the robustness against postural perturbations is two times higher in the nmF model compared to the FMCH model and even further increased in the adaptive nmF model. This is due to the intrinsic muscle dynamics and the tuning of the reflex gains. With this, we demonstrate, for the first time, the evolution of mechanical template models (e.g. VP concept) to a more physiological level (nmF model). This shows that the template model can be successfully used to design and control robust locomotor systems with more realistic system behaviours.

# 1. Introduction

Building a legged system with agile, efficient and robust functionality requires appropriate design and control. To this end, roboticists can learn from biological locomotor systems. As a result, the biological body as the representative of the mechanical design and the neural system as the controller can be considered for understanding the basic concepts of legged locomotion. However, detailed modelling of such an intricate phenomenon is challenging and also computationally expensive. Employing template (conceptual) models were introduced as an alternative method for realizing the fundamentals of locomotion [1]. The striking feature of templates is that they ignore all the redundancies and still lend themselves to be used as simple conceptual models explaining complex problems [2].

As one of such templates, the spring-loaded inverted pendulum (SLIP), which is composed of a massless spring along with a point mass at top, can remarkably describe human running [3,4] and walking [5,6]. This simple conceptual model also helped develop some legged robots in the past [7,8]. For addressing human posture control, the extension of the point mass with a rigid trunk was introduced in TSLIP [9] or ASLIP [10] for running and BTSLIP (bipedal TSLIP) [11,12] for walking. In constructing a model that resembles the salient features of human locomotion, employing proper control techniques for balancing the upper body warrants consideration. Several methods comprising hip joint control have been proposed for stabilizing the upper body [7,9,12–14]. From another perspective, bioinspired control concepts such as virtual pendulum (VP) [11] were introduced based on analysing and modelling human (or animal) gaits. Therefore, inspiration from nature (e.g. in virtual pendulum posture control (VPPC [9]) or force modulated compliant hip (FMCH [12])) may open new doors in the design and control of legged systems which might not be accessible by engineering approaches. In FMCH, the leg force is employed to adjust hip compliance and to generate the required hip torque for balance control accordingly. This method can stabilize running [14] and walking [12] while predicting human hip torques in walking [15]. Furthermore, this method presents a simple physical representation for implementing the control concept using minimal exchange of sensory information between different locomotor subfunctions (stance and balance) [15].

Our selected road-map for the evolution of models from template to anchor and the position of the current study are shown in figure 1. Here, we take a step towards developing a biologically plausible version of the VPP concept (figure 1$d$). For this, we extend the FMCH to the neuromuscular level by replacing hip springs with the Hill-type muscle model [16]. This way, the leg force plays the role of a reflex signal to activate hip muscles. The nmF (neuromuscular FMCH) and FMCH are compared regarding their abilities in predicting human walking data. Furthermore, to deal with external disturbances, we design a discrete linear quadratic regulator (LQR) using Poincaré map to adapt the muscle feedback gains in an event-based manner. The robustness and efficiency of the proposed methods (nmF and adaptive nmF) are analysed and compared to those of the FMCH model and a feed-forward control approach. This latter approach which mimics preflex control [17] in the human body is considered as a baseline for stabilizing property of a controller without feedback. This minimalistic feed-forward control showed already acceptable quality in stabilizing bipedal robot gaits [18].

The remainder of the paper is organized as follows: §2 describes the development and formulation of the presented neuromuscular model and the developed control approaches. In §3, the walking results (from human experiments and modelling) are presented and the robustness of the proposed approaches against external perturbations is investigated by demonstrating the basin of attraction for each method. The discussion about the results and predictability of the VPP concept by different models besides the future of this research are described in §4. Finally, §5 concludes the paper. The abbreviations used throughout the paper are summarized in table 1.

# 2. Methodology

In this section, we explain the new nmF (neuromuscular FMCH) model which is used for posture control in walking. This model is placed between template models and neuromuscular anchor models (figure 1). In addition, the event-based control for adapting the reflex gains to increase robustness against uncertainties and perturbations is presented.

## 2.1. Modelling

The model adopted for simulations in this paper is an extension of the bipedal TSLIP (BTSLIP) model in the sagittal plane equipped with hip muscles (figure 2). Combining this model with the FMCH control

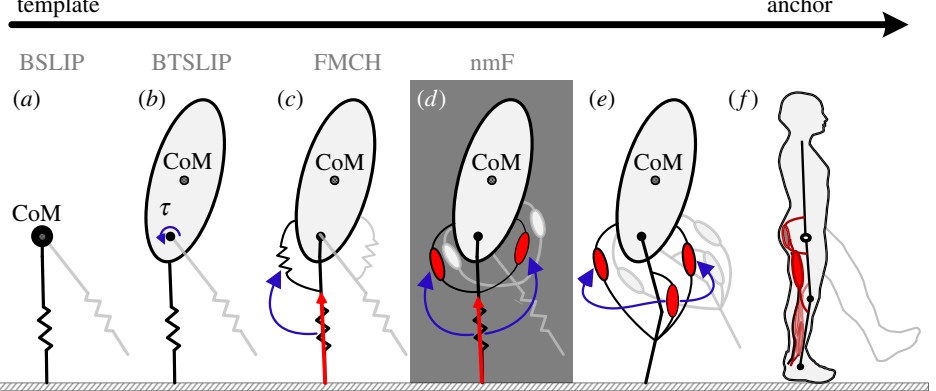

**Figure 1.** Model evolution from template to anchor; (*a*) BSLIP template with a point mass at the hip joint and two massless springs for the legs, (*b*) BSLIP model extended by substituting the point mass with a rigid trunk; this model is also known as BTSLIP, (*c*) FMCH model wherein leg forces are used as feedback signals to generate required hip torques for posture control, (*d*) nmF model investigated in this paper, where hip springs are replaced by Hill-type muscle models, (*e*) the next step of modelling by adding segmented legs and biarticular muscles, and (*f*) a complete neuromuscular (anchor) model.

**Table 1.** Nomenclature.

| | abbreviation | definition |
|---|---|---|
| | CoM | centre of mass |
| | BW | body weight |
| | GRF | ground reaction force |
| | PTS | preferred transition speed |
| | SLIP | spring loaded inverted pendulum |
| | TSLIP | SLIP extended by trunk |
| | BTSLIP | bipedal TSLIP |
| | FMCH | force modulated compliant hip |
| | nmF | neuromuscular FMCH |
| | VBLA | velocity-based leg adjustment |
| | VPPC | virtual pendulum posture control |
| | VP | virtual pendulum |
| | LQR | linear quadratic regulator |
| | RF | rectus femoris |
| | HAM | hamstrings |
| | ECC | excitation contraction coupling |
| | MTC | muscle tendon complex |
| | CE | contractile element |
| | SEE | series elastic element |
| | MCx | motor cortex |

concept results in the nmF (neuromuscular FMCH) model. In the following, we explain both the BTSLIP model and the muscle-tendon unit modelling.

### 2.1.1. BTSLIP model

An extension of the SLIP model (figure 1*a*) for bipedal walking including the upper body to address posture control was introduced in [11]. This BTSLIP model, which is shown in figure 1*b*, comprises a rigid trunk that

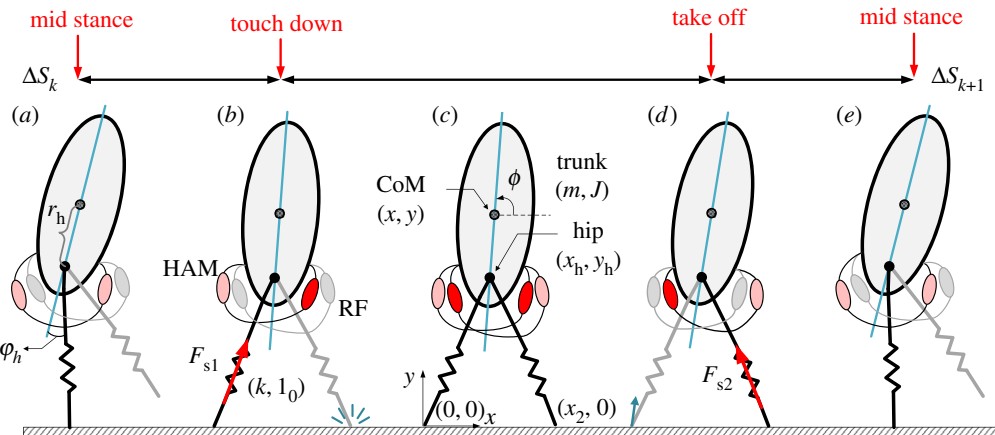

**Figure 2.** (a−e) The neuromuscular BTSLIP model with a rigid trunk and legs modelled as massless prismatic springs and definitions of walking phases. A rectus femoris (RF) and a hamstrings (HAM) muscle group are added to each leg. The muscles involved during each phase of walking are coloured in red with different shades. Muscles with dark red colour inject more force than those shown in light red. Also, the muscles silent in each phase are shown in grey. The chequered circles approximate the centre of mass (CoM) location.

travels on two massless compliant legs. Similar to the SLIP, in the BTSLIP model, the segmented leg is represented by a virtual leg, between the hip and the foot. In the FMCH model (figure 1c), adjustable springs and in the nmF model, two muscles between each leg and the trunk (figure 1d) provide the required hip torques. Except the hip actuation mechanism, the rest of these three models are equal, as explained in the following. In figure 2, a schematic of the model is illustrated in one step.

The walking gait comprises two phases: single support and double support. The single support phase describes a period in which only one leg is in contact with the ground (stance leg) and the other, called swing leg, moves forward until its distal end hits the ground (i.e. touch down). Following the onset of swing leg contacting the ground, the double support phase (having two stance legs) starts. At the end of this phase, the former stance leg takes off, while the former swing leg remains on the ground which results in the next single support phase. In both phases, the hip torques exerted between the trunk and the stance legs ($\tau_1$, $\tau_2$) control the interactions between the stance leg and the upper body.

Changes between single support and double support phases can be easily identified as one leg touches or leaves the ground. For a leg, the stance phase initiates once the leg touches the ground. During this phase, the spring force along the leg axis is given by $F_s = k(l - l_0)$ where, $l$, $l_0$ and $k$ denote the leg length, the rest length of leg spring and stiffness, respectively. Stance phase ends once the leg leaves the ground. This moment is detected when the ground reaction force (GRF) has no vertical component. That is when $GRF_y = 0$. According to figure 2, the generalized coordinates required for describing the dynamical equations of the BTSLIP model are chosen as $[x, y, \phi]^T$. In this vector, $x$, $y$ and $\phi$ represent the horizontal and vertical positions of CoM and trunk orientation, respectively, while the superscript $^T$ denotes transpose function, hereafter. As shown in figure 2, the hip position ($X_h = [x_h, y_h]^T$), which is the intersection of the two legs on the trunk, is placed below the CoM with distance $r_h$:

$$\left.\begin{aligned} x_h &= x - r_h \cos \phi \\ y_h &= y - r_h \sin \phi. \end{aligned}\right\} \quad (2.1)$$

and

Also, the ground reaction force $GRF = [GRF_x, GRF_y]^T$ is produced as a result of the hip torque $\tau$ and the spring force $F_s$ along the leg axis:

$$\left.\begin{aligned} GRF_x &= F_s \frac{x_h}{l} + \tau \frac{y_h}{l^2} \\ GRF_y &= F_s \frac{y_h}{l} - \tau \frac{x_h}{l^2}. \end{aligned}\right\} \quad (2.2)$$

and

The dynamical equations in the single support phase can be written as follows:

$$\left.\begin{aligned} m\ddot{x} &= GRF_x, \\ m\ddot{y} &= GRF_y - mg \\ J\ddot{\phi} &= \tau + r_h(GRF_x \sin \phi - GRF_y \cos \phi), \end{aligned}\right\} \quad (2.3)$$

and

where $g$ is the gravitational acceleration and $m$ is the body mass.

Deriving the motion dynamics in the double support phase amounts to taking into account the spring force of both legs. For the sake of simplicity, unless otherwise specified, the parameters related to the first and second legs are denoted by subscripts $_1$ and $_2$, respectively. In this phase, the controller applies hip torques ($\tau_1, \tau_2$) to maintain balance and consequently to keep the system stable. Defining the position of the second stance leg by $[x_2, 0]$,[1] the dynamical equations in the double support phase are as follows:

$$
\left.
\begin{aligned}
m\ddot{x} &= \mathrm{GRF}_{x_1} + \mathrm{GRF}_{x_2}, \\
m\ddot{y} &= \mathrm{GRF}_{y_1} + \mathrm{GRF}_{y_2} - mg, \\
J\ddot{\phi} &= \tau_1 + \tau_2 + r_\mathrm{h}(\,\mathrm{GRF}_{x_1} + \mathrm{GRF}_{x_2})\sin\phi \\
&\quad - r_\mathrm{h}(\,\mathrm{GRF}_{y_1} + \mathrm{GRF}_{y_2})\cos\phi,
\end{aligned}
\right\}
$$
(2.4)

and

where

$$
\left.
\begin{aligned}
\mathrm{GRF}_{x1} &= F_{\mathrm{s}1}\frac{x_\mathrm{h}}{l_1} + \tau_1\frac{y_\mathrm{h}}{l_1^2}, \\
\mathrm{GRF}_{y1} &= F_{\mathrm{s}1}\frac{y_\mathrm{h}}{l_1} - \tau_1\frac{x_\mathrm{h}}{l_1^2}, \\
\mathrm{GRF}_{x2} &= F_{\mathrm{s}2}\frac{x_\mathrm{h} - x_2}{l_2} + \tau_2\frac{y_\mathrm{h}}{l_2^2}, \\
\mathrm{GRF}_{y2} &= F_{\mathrm{s}2}\frac{y_\mathrm{h}}{l_2} - \tau_2\frac{x_\mathrm{h} - x_2}{l_2^2}
\end{aligned}
\right\}
$$
(2.5)

and

$$
l_i = \sqrt{(x_i - x_\mathrm{h})^2 + (y_i - y_\mathrm{h})^2}, \quad i = 1, 2.
$$
(2.6)

### 2.1.2. Muscle model

In the nmF model, we replace the adjustable hip springs of the FMCH model [12] with the Hill-type muscle models for posture control. Here, we briefly describe the muscle modelling. In the proposed neuromuscular model which is added on the top of the BTSLIP model, the hip torques ($\tau_1, \tau_2$) result from muscle forces. Each leg has its own pair of muscles. In general, the muscle-tendon complex (MTC) consists of a contractile element (CE) and a series elastic element (SEE). However, we only consider the CE part of the MTC to investigate the muscle properties (similar to [19]). In the Hill-type muscle model [16], the generated force by CE is given by

$$
F_\mathrm{CE}(A, l_\mathrm{CE}, v_\mathrm{CE}) = AF_\mathrm{max}f_\mathrm{l}(l_\mathrm{CE})f_\mathrm{v}(v_\mathrm{CE}),
$$
(2.7)

where $A$ is the muscle activation level which accepts values between 0 and 1. In addition, $l_\mathrm{CE}$, $v_\mathrm{CE}$ and $F_\mathrm{max}$ are the muscle length, contraction speed and maximum isometric force, respectively. For the force–length $f_\mathrm{l}(l_\mathrm{CE})$ and the force–velocity $f_\mathrm{v}(v_\mathrm{CE})$ relations, we use the equations introduced by Aubert [20], which are used in the neuromuscular model of Geyer et al. [21–23].

$$
f_\mathrm{l}(l_\mathrm{CE}) = \exp\left[ c\left|\frac{l_\mathrm{CE} - l_\mathrm{opt}}{l_\mathrm{opt}w}\right|^3 \right].
$$
(2.8)

In this equation, $l_\mathrm{opt}$ is the optimal CE length, $w$ denotes the width of the bell-shaped $f_\mathrm{l}(l_\mathrm{CE})$ curve and $c$ is a constant value that can be considered as the muscle stiffness. The force–velocity $f_\mathrm{v}(v_\mathrm{CE})$ is as follows:

$$
f_\mathrm{v}(v_\mathrm{CE}) = \begin{cases} \frac{v_\mathrm{max} - v_\mathrm{CE}}{v_\mathrm{max} + \kappa v_\mathrm{CE}} & v_\mathrm{CE} < 0 \\ N + (N-1)\frac{v_\mathrm{max} + v_\mathrm{CE}}{7.56\kappa v_\mathrm{CE} - v_\mathrm{max}} & v_\mathrm{CE} \geq 0 \end{cases},
$$
(2.9)

where, $v_\mathrm{max}$, $\kappa$ and $N$ are constant values representing the maximum contraction velocity, the curvature constant and the eccentric force enhancement [20], respectively. The values for all the constant coefficients are borrowed from [21].

The antagonistic pair of muscles in the nmF model represent the muscles actuating the upper limb of the human segmented leg. The biarticular thigh muscles (i.e. rectus femoris (RF) and hamstrings (HAM)) in the human body contribute to both knee and hip joints. In [24], it was shown that by setting the hip to

---

[1]It is assumed that the rear leg is positioned at [0, 0].

knee lever arm ratio of a biarticular thigh muscle to 2, the segmented leg model, shown in figure 1e, will be equal to the nmF model, depicted in figure 1d. Hence, by representing the segmented leg with a virtual compliant leg, the biarticular thigh muscles can be responsible for balance control without cross-talk to the stance leg axial direction. Now, according to figure 2, the RF and HAM muscle lengths ($L_{RF}$ and $L_{HAM}$) are related to the angle between the trunk and the corresponding leg ($\varphi_h$). In the following, the equations for the single support or the rear leg in the double support are presented. Considering the position of the foot contact, similar formulation can be found for the front leg.

$$\left.\begin{array}{l} L_{RF} = L_{RF_0} + \rho r_0(\varphi_h - \varphi_{ref(RF)}) \\ L_{HAM} = L_{HAM_0} + \rho r_0(\varphi_{ref(HAM)} - \varphi_h) \end{array}\right\} \qquad (2.10)$$

and

in which, $r_0$ is the constant moment arm of the muscle and $L_{RF_0}$ and $L_{HAM_0}$ represent the rest length of the RF and HAM muscles, respectively. Also, $\rho$ accounts for the muscle pennation angles and $\varphi_{ref(RF)}$ and $\varphi_{ref(HAM)}$ are the reference joint angles defined for the RF and HAM muscles, respectively. In addition, the hip angle can be calculated as follows:

$$\varphi_h = \phi - \tan^{-1}\left(\frac{y_h}{x_h}\right). \qquad (2.11)$$

The torque that each muscle (RF and HAM) exerts at the hip joint for one leg is given by

$$\left.\begin{array}{l} \tau_{RF} = F_{CE}(L_{RF})r_0 \\ \tau_{HAM} = F_{CE}(L_{HAM})r_0 \end{array}\right\} \qquad (2.12)$$

and

and the total hip torque is resulted from these two torques as

$$\tau = \tau_{HAM} - \tau_{RF}. \qquad (2.13)$$

As already mentioned, in the double support phase, similar torques can be found for the second leg that should be summed up with this value to calculate the total hip torque on the upper body. Using these equations for the hip torque in equations (2.2)–(2.5) completes the dynamical formulation of the neuromuscular BTSLIP model. The next step is to address the control approach required for achieving stable and robust walking.

## 2.2. Control approach

Inspired by the Raibert control approach [7], we consider three locomotor subfunctions [25] to generate a stable gait: (i) stance, the axial function of the stance leg; (ii) swing, the rotational movement of the swing leg; and (iii) balance, upper body posture control. Since the stance leg is passive (linear spring), the control strategy for the BTSLIP model consists of the swing and balance subfunctions (for more details about the locomotion subfunction concept see [25]). In the swing phase, a leg placement strategy for adjusting the swing leg orientation is needed. For this, the leg angle of attack should result in appropriate initiation of the next double support phase. The second task is stabilization of the upper body which is exerted in both stance and swing phases. Accordingly, we propose an adaptive control approach to maintain balancing. In the following, we detail our proposed control methods at each phase for achieving a stable and robust walking.

Stability is defined based on the step to fall concept which is a common approach in model-based model analyses of legged locomotion [26,27]. Here, the model can predict stable walking at a certain speed if it can take 50 steps, while the variations in motion speed at each step are not more than 5% of the specified speed. In other words, stability is defined by staying in the neighbourhood of a limit cycle. The system is robust against a certain perturbation, if it can maintain stability, specified here as 50 steps after perturbation occurrence.

### 2.2.1. Velocity-based leg adjustment during swing phase

A fixed angle of attack (swing leg angle with respect to ground) is sufficient for stable running [26] and walking [5]. However, robustness against perturbations or uncertainties with a large domain of attraction cannot be achieved without leg angle adaptation. In earlier studies, the adjustment of the leg orientation was accomplished based on the horizontal velocity [10,28,29]. In [30], a robust strategy for controlling the leg angle, named VBLA (velocity-based leg adjustment) is proposed which takes into account both the

CoM velocity and the gravity vectors. The potential to mimic human leg adjustment in different gaits, supporting larger range of walking velocities and having higher robustness against perturbations are some of the advantages of VBLA compared to previous approaches [30]. Due to these characteristics, we select this method in this paper.

In VBLA, a weighted average of the CoM velocity vector $V$ and gravity vector $G$ yields the leg direction $O$ as follows:

$$O = (1 - \mu)V + \mu G, \tag{2.14}$$

with weighting constant $\mu$ ranging from 0 to 1. Here, $\mu = 0$ corresponds to a position where the leg is in parallel with the CoM velocity vector and $\mu = 1$ results in vertical leg orientation.

## 2.2.2. Posture stabilization with neuromuscular FMCH

An efficient stable bipedal gait (e.g. in humans) requires an upright posture [11]. To generate stable walking with the neuromuscular BTSLIP model, a proper control framework for balancing the upper body is needed. In [12], the leg FMCH was introduced as a new model for human-like postural control in walking. In this work, leg forces were used as feedback signals to exert the torque profile required at the hip joint. This concept was then applied to a BTSLIP model through stiffness adjustment of the rotational springs for stabilizing the posture (figure 1c). In the FMCH model, the hip torques are given with the following equation:

$$\tau = GF_s(\varphi_{h0} - \varphi_h) \tag{2.15}$$

in which, $G$, $F_s$, $\varphi_h$ and $\varphi_{h0}$ are the normalized stiffness, the leg force, the angle between the upper body and the leg and the rest angle of the spring, respectively. It was shown that the leg force feedback (e.g. approximated using sensory information from knee extensor muscles) is crucial to tune the hip compliance, required for balancing. This model can generate different stable gaits (hopping, running [14] and walking [12]) in addition to precise prediction of human posture control in walking at different speeds [15].

As an extension of FMCH to the neuromuscular level, in this work, the leg force $F_s$, which is the GRF in the leg axial direction, is employed as a sensory pathway for activating RF and HAM muscles. For that, the sensory signal ($F_s$) is delayed, gained and subsequently passed through the excitation-contraction coupling (ECC) to create the muscle activation $A$ (used in Hill muscle model, equation (2.7)). A similar reflex scheme was shown to be advantageous in generating a stable hopping when using muscle length and force feedbacks [21]. However, it differs from our approach in the sense that we employ the leg force instead of muscle force as the sensory feedback. As the leg force $F_s$ can be approximated by vastus muscle force in human body, our control approach can be interpreted as using the reflex signal from another muscle (vastus) to adjust the activation of the biarticular thigh muscles. Similar reflex pathways are used for walking in two dimensions [22] and three dimensions [23], while the control structure and feedback gains are obtained by optimization methods. Instead, our reflex control is developed based on a bioinspired posture control method (VPP concept). Figure 3 illustrates the proposed control structure, schematically. This model is termed nmF, standing for neuromuscular FMCH, which is demonstrated in figure 3 within the green dashed box. This neural feedback (reflex) can be formulated as follows:

$$\left.\begin{array}{c} \text{STIM}(t) = \text{STIM0} + GF_s(t - \Delta P) \\[2mm] T\dfrac{\partial A}{\partial t} = \text{Sat}(\text{STIM}(t)) - A(t) \end{array}\right\} \tag{2.16}$$

and

where in the first line, STIM, STIM0 and $\Delta P$ are defined as the stimulation signal, the stimulation bias and the feedback delay, respectively. In the second line, the Sat function is for saturating the stimulation to a predefined range and a first-order differential equation relating stimulation to activation signal is described with $T$ being a time constant.

To demonstrate the different levels of reaction to perturbations, we compare *preflex control* [31] with adaptive nmF. Preflexes are defined as the intrinsic properties of muscles at the lowest level of the hierarchical sensorimotor neuromuscular control [17]. An optimal predefined stimulation to control muscle activation exhibits similarities to human preflex behaviour [31] with minimal effort for generating repetitive motions [18,32]. In this feed-forward control, the activation signals to the muscles are chosen as those obtained in the nmF model in steady walking. Finally, in adaptive nmF, for

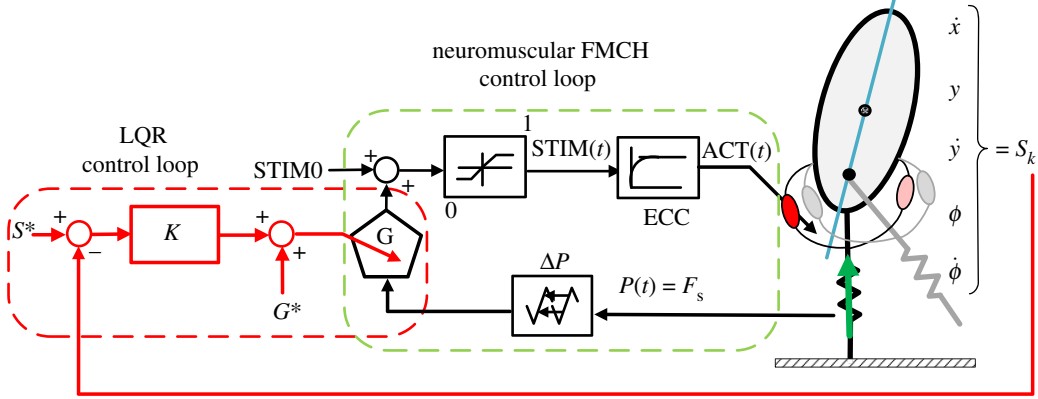

**Figure 3.** Control loop schematic. In the inner control loop (i.e. neuromuscular FMCH control loop), the stance leg force is used as the feedback signal for creating muscle activations. In the outer loop, shown in red (i.e. LQR control loop), a discrete LQR controller is implemented to tune the reflex gains once per step at mid-stance (even based).

increasing the robustness against perturbations we employ a discrete LQR controller to readjust the muscle gains at each step. More details are discussed in the following.

### 2.2.3. Adaptive nmF for posture control

Our proposed neuromechanical model addresses balance control in bipedal walking. Using reflex control in the nmF model provides robustness against perturbations based on low-level sensory information and motor control. However, in human locomotion, higher level control (e.g. from brain based on visual or vestibular system) can increase robustness by reacting to larger perturbations. For implementing such a higher control layer, we adapt the reflex gains using a discrete LQR (linear quadratic regulator) [33], once per step. We use $S = [\dot{x}_k, y_k, \dot{y}_k, \phi_k, \dot{\phi}_k]$ as the descritized model of walking by linearizing the continuous system dynamics about the Poincaré section $S^*$. The mid-stance configuration in which the stance leg is completely vertical in the middle of the single support is used to define $S^*$. A stable limit cycle in the continuous model equals the stable fixed point of the discrete model. In our reduced order linearized model, the horizontal position of the CoM ($x_k$) is skipped, because this state is incremental during forward walking and its regulation to a fixed value means stopping movement which is against stable walking at a certain speed. The discrete linearized model is given by

$$\Delta S_{k+1} = A_s \Delta S_k + B_s \Delta U_k, \tag{2.17}$$

with

$$\Delta S_k = S_k - S^* = [\Delta \dot{x}_k, \Delta y_k, \Delta \dot{y}_k, \Delta \phi_k, \Delta \dot{\phi}_k], \tag{2.18}$$

$$\Delta U_k = U_k - U^* = [G_{HAM} \quad G_{RF}]_k^T - [G_{HAM}^* \quad G_{RF}^*]^T \tag{2.19}$$

and

$$A_s = \frac{\partial F}{\partial S}(S^*, U^*), \quad B_s = \frac{\partial F}{\partial U}(S^*, U^*), \tag{2.20}$$

where $F$ represents the dynamical equation of the system and the Jacobian matrices $A_s$ and $B_s$ are the derivatives of $F$ with respect to the state and the input vectors at $S^*$, respectively. Here, $U_k$ denotes the control vector which comprises RF and HAM muscle feedback gains. Also, $U^* = G^* = [G_{HAM}^* \quad G_{RF}^*]^T$ indicates the initial muscle gains before adaptation.

Since the stability of the original continuous system is related to the stability of the corresponding Poincaré map, we employ a discrete LQR controller which minimizes the following cost function to stabilize the discrete dynamical model of equation (2.17):

$$J = \sum_{k=1}^{\infty} \Delta S_k^T Q \Delta S_k + \Delta U_k^T R \Delta U_k, \tag{2.21}$$

where $Q$ and $R$ are weighting factors defined according to performance and stability metrics. The optimal control sequence minimizing equation (2.21) and stabilizing equation (2.17) is given by state feedback

$$\Delta U_k = -K \Delta S_k, \tag{2.22}$$

**Table 2.** BTSLIP model parameters. The model parameters are set to match the characteristics of an ordinary human being.

| parameter | definition | value (units) |
|---|---|---|
| $m$ | trunk mass | 80 (kg) |
| $J$ | trunk moment of inertia | 4.6 (kg m$^2$) |
| $r_h$ | distance hip–CoM | 0.1 (m) |
| $L_0$ | leg rest length | 1 (m) |
| $g$ | gravitational acceleration | 9.81 (m s$^{-2}$) |
| $\mu$ | leg adjustment parameter | 0.34 |
| $k_N$ | normalized leg stiffness | 40 |

with $K$ being the optimal gain vector found iteratively from the following equations:

$$K = (B_s^T P B_s + R)^{-1} B_s^T P A_s \tag{2.23}$$

and

$$P = Q + A_s^T (P - P B_s (B_s^T P B_s + R)^{-1} B_s^T P) A_s \tag{2.24}$$

In this paper, we refer to the combination of neuromuscular FMCH and LQR controller as *adaptive nmF*. The control loop schematic is illustrated in figure 3, by the red dashed line.

## 2.3. Human experiment

In order to compare the modelling results with human experiments, we have employed a dataset published in [34]. This dataset was collected in walking experiments on a treadmill (type ADAL-WR, Hef Tecmachine, Andrezieux Boutheon, France) at different speeds. However, we have selected one speed for comparison. For other speeds, we need to tune the model parameters if it is in the achievable walking speeds with SLIP-based models (i.e. 0.5–1.5 m s$^{-1}$). Motion capture data and GRF data are measured by Qualisys set-up (Gothenburg, Sweden) from 11 markers and from force sensors within the treadmill, respectively. Twenty-one subjects (11 females, 10 males, age: 22–28 yrs, height: 1.64–1.82 m, weight: 59.2–82.6 kg) were asked to walk at different percentages of their preferred transition speeds (PTS).[2] The treadmill speed was employed as the walking speed. Here, we use walking data for 50% PTS which is about 1 m s$^{-1}$, addressing *moderate walking speed*. Kinematic and kinetic data processing were described in [34].

# 3. Results

In this section, the performance of the proposed control schemes on the neuromuscular BTSLIP models is compared in walking. We use the VBLA for leg adjustment and force feedback-based strategies by modulating the hip spring stiffness (FMCH) or muscle activation (nmF and adaptive nmF) for balance control. The BTSLIP and nmF model parameter values are summarized in tables 2 and 3, respectively. The forward velocity considered for this model is 1 m s$^{-1}$. Also, unless noted otherwise, the values of $Q$ and $R$ matrices are chosen as identity matrices of proper dimensions, giving similar weights to the error and energy consumption.

## 3.1. Comparison to human walking

Figure 4 shows the simulation and experimental results from different perspectives (kinetics and kinematics). In figure 4a, the CoM movement in vertical direction is depicted for FMCH and nmF models along with the human data. Double humped vertical displacement of CoM during one stride of walking is observed in both human experiments and simulation models. The CoM vertical displacements are about 5 cm for both cases. Figure 4b illustrates the torque applied by the hip springs for FMCH model and by the hip muscles for nmF model. Except the time shift seen in this figure, one can easily observe that the torque profile of the model is the same as the human hip

---

[2]PTS is the preferred speed for transition between running and walking which is typically about 1.9–2.1 m s$^{-1}$ for humans [34].

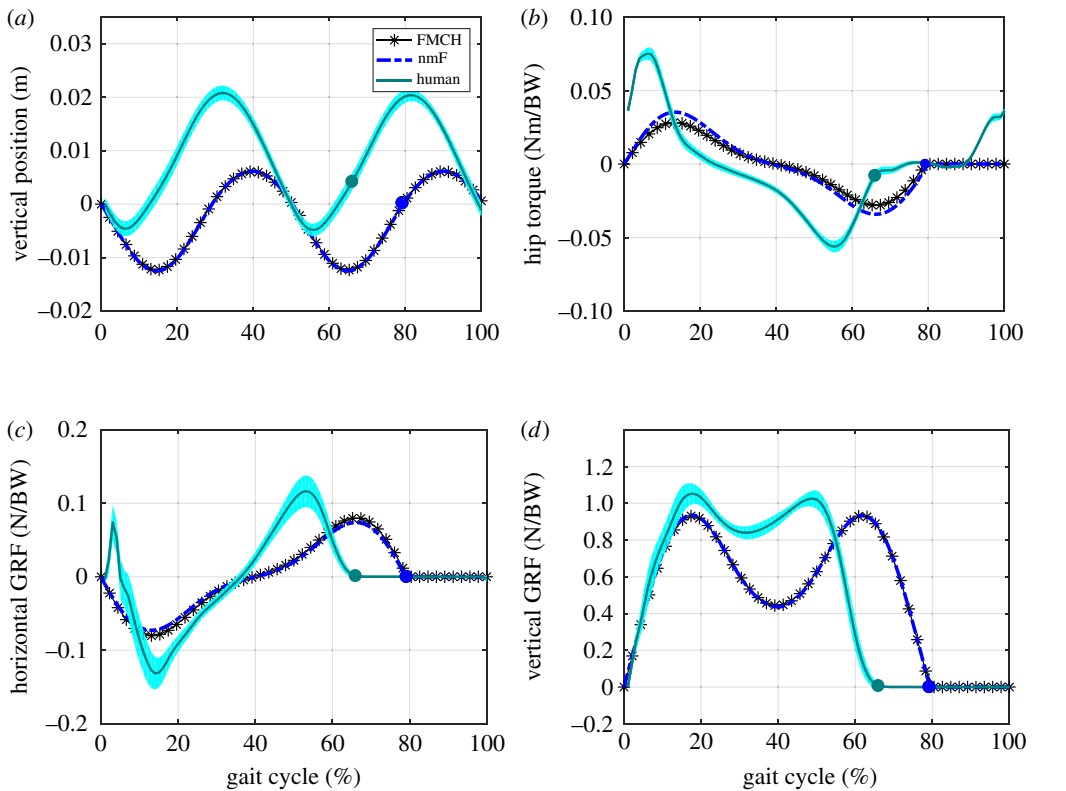

**Figure 4.** Comparison between the FMCH model, nmF model and human experimental results in one stride (two steps) at 1 m s$^{-1}$ walking speed. Each stride starts at touchdown of one leg to the next touchdown of the same leg. Dots shown in the figures signify the toe-off moment. The cyan solid lines and the shaded areas surrounding them show the mean and variance for 21 subjects, respectively. The black lines with star markers illustrate the graphs for FMCH and the blue dashed lines depict the results for nmF model. (a) Vertical displacement of the CoM from the initial height at touchdown, (b) hip torque variations normalized to the body weight (BW), (c,d) horizontal and vertical GRFs normalized to BW.

**Table 3.** Muscle model parameters.

| parameter | definition | value (units) | |
|---|---|---|---|
| | | RF | HAM |
| $L_{mtu0}$ | MTU reference length | 0.11 (m) | 0.11 (m) |
| $\phi_{ref}$ | reference joint angle | 3.291 (rad) | 3.246 (rad) |
| $r$ | constant lever contribution | 0.1 (m) | 0.1 (m) |
| $F_{max}$ | maximum isometric force | 2000 (N) | 2000 (N) |
| $L_{opt}$ | optimum length | 0.111 (m) | 0.111 (m) |
| $V_{max}$ | maximum shortening velocity | $-12(L_{opt}s^{-1})$ | $-12(L_{opt}s^{-1})$ |
| $\tau$ | excitation-contraction coupling | 1 (ms) | 1 (ms) |
| $\Delta p$ | feedback time delay | 1 (ms) | 1 (ms) |
| STIM0 | constant value | 0.01 | 0.01 |
| $G$ | gain of muscle feedback | $0.624/F_{max}$ | $0.936/F_{max}$ |
| $w$ | width of bell-shaped | 0.2 (m) | 0.2 (m) |
| $c$ | constant value | $\ln(0.05)$ | $\ln(0.05)$ |
| $N$ | eccentric force enhancement | 1.5 | 1.5 |
| $\kappa$ | curvature constant | 5 | 5 |
| $\rho$ | muscle pennation angle | 0.5 | 0.5 |

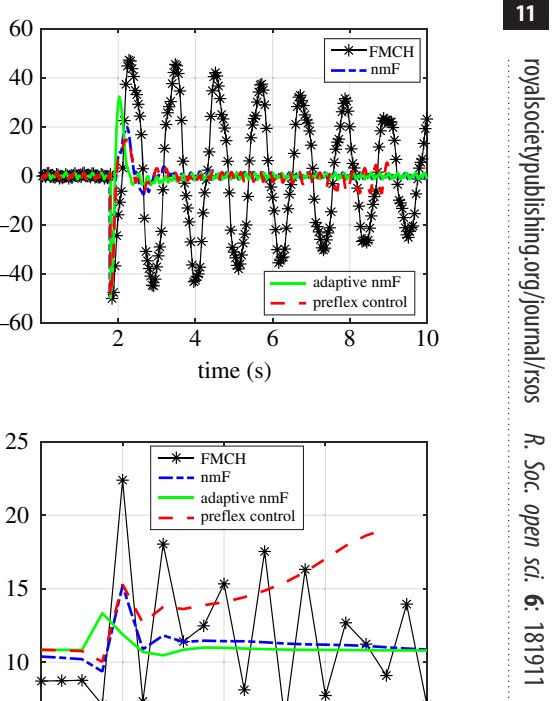

**Figure 5.** Simulation results showing the performance of different models in face of an external perturbation. (*a*,*b*) Orientation and angular velocity of trunk in the presence of an external perturbation at the beginning of fourth stride, (*c*) variation of RF and HAM muscle feedback gains with respect to number of strides in response to perturbation for nmF and adaptive nmF models and (*d*) consumed energy at the hip joint in different strides.

torque profile. Figure 4*c*,*d* shows the horizontal and vertical components of the GRF. The results from the horizontal GRF depict similarity to what is observed in humans. The small positive peak force at the onset of walking cycle in human data shows the impact effect of touchdown which is missing in our models because the legs are massless. This simplification could also result in shorter swing phase in SLIP-based models compared to human walking. These differences could also be attributed to the absence of foot in the BTSLIP model, but still the patterns and the GRF magnitudes are comparable.

## 3.2. Robustness against external perturbations

To investigate the robustness of the model along with the employed control approach in face of external disturbances, we perturbed the system by a sudden push on the upper body. For emulating this impact, the trunk angular velocity ($\dot{\phi}$) is largely magnified (to $-50°$), instantaneously. The results of comparing the preflex, the FMCH, the nmF and the adaptive nmF control methods, are displayed in figure 5. The orientation of the trunk with respect to the horizontal axis and its angular velocity are depicted in figure 5*a* and 5*b*, respectively. In the preflex control, after the perturbation, the trunk is leaning forward and its angle is deviating from the vertical orientation. This situation is perpetuated for almost 7 s until both legs lose contact with the ground and accordingly the model loses its stability. In the FMCH model, large oscillations in the trunk orientation are induced due to the perturbation occurrence. Yet, as seen in the figures, this model is able to recover from the perturbation and continues walking. In the nmF model, the trunk angle tends to return to the upright posture more quickly and it takes less than 2 s for this model to return to its previous orientation pattern. This is a significant improvement in terms of perturbation recovery compared to the aforementioned models. Adaptation of the control gains with discrete LQR also enables the body to keep upright posture after perturbation occurrence with a settling time similar to nmF, without considerable oscillations. The feedback gains for both nmF and adaptive nmF models in course of convergence to the steady walking are illustrated in figure 5*c*. This figure shows that following the onset of perturbation, the

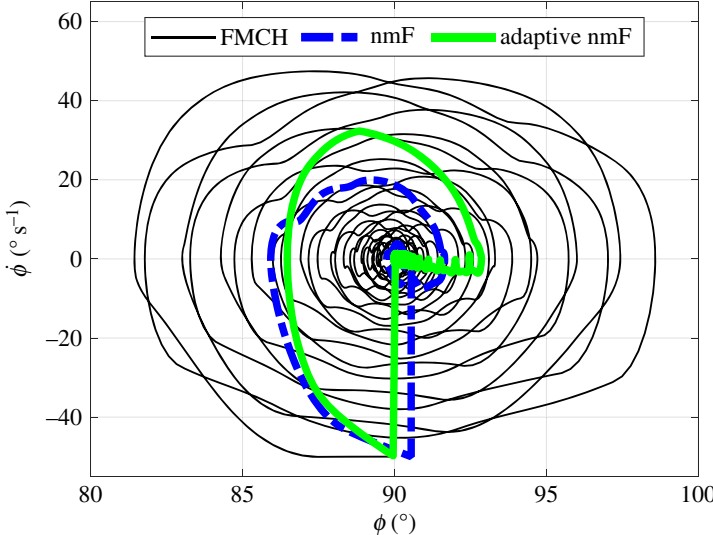

**Figure 6.** Phase portrait of the trunk angle ($\phi$) for the perturbed system controlled with FMCH, nmF and adaptive nmF methods.

muscle gains are adjusted and then settled quickly to their previous values. Finally, figure 5d displays the consumed energy J at each stride for all models. Here, the consumed energy in one stride (with duration T) is computed from multiplication of hip torque $\tau_i$ and hip angle $\varphi_{h_i}$, ($i = 1, 2$) as follows:

$$J = \int_0^{\mathrm{T}} |\dot{\varphi}_{h_1}(t)\tau_1(t)| + |\dot{\varphi}_{h_2}(t)\tau_2(t)| \, \mathrm{d}t, \tag{3.1}$$

in which the absolute value function is used to sum up the magnitude of power from the positive and negative works. Thanks to the inherent characteristics of nmF model, less power and consequently energy is consumed during rejecting the disturbance and converging to a stable walking gait compared to the FMCH model. The results for the adaptive nmF model also show that adjusting reflex gains according to the LQR controller output has effectively helped reduce the consumed energy after perturbations.

To better understand the dynamic behaviour in reaction to perturbations, the phase portrait (with $\phi$ and $\dot{\phi}$ being the state variables) is displayed in figure 6 for all models except preflex control, which is unstable after perturbation. Starting from similar initial conditions, the states in the (adaptive) nmF model converge to the stable limit cycle much faster than those of the FMCH model. Since a compromise between efficiency and performance can be set by tuning R and Q in the LQR method (adaptive nmF), increasing efficiency (figure 5d) results in lower performance (higher deviation from the limit cycle, figure 6) compared to nmF, which is not significant.

## 3.3. Efficiency and performance in adaptive nmF

The constraints and potentials of the proposed adaptive method can be verified by investigating the effect of weighting R and Q on the efficiency and performance of locomotion. By now, we always set these matrices to identity matrix I. Here, we repeat the perturbation recovery simulations for three different cases: (i) $Q = 100I$, $R = I$, (ii) $Q = R = I$ and (iii) $Q = I$, $R = 100I$. The results obtained from this simulation are illustrated in figure 7. The effects of selecting different pairs of (Q, R) on the orientation of trunk and its angular velocity are depicted in figure 7a and 7b, respectively. The highest performance with the fastest regulation to steady state after perturbation is achieved in case (i), in which Q is larger than R. Also, by calculating the peak power and energy consumption at the hip joint as shown in figure 7c and 7d, respectively, it can be observed that case (i) has the highest of both. Putting more weight on the R matrix as in case (iii) yields the lowest control effort (energy and peak power) while having the largest oscillations after perturbations.

## 3.4. Basin of attraction

Robustness against perturbations among different methods are compared in figures 5 and 6, by investigating the response to a sample perturbation. For a more comprehensive and descriptive

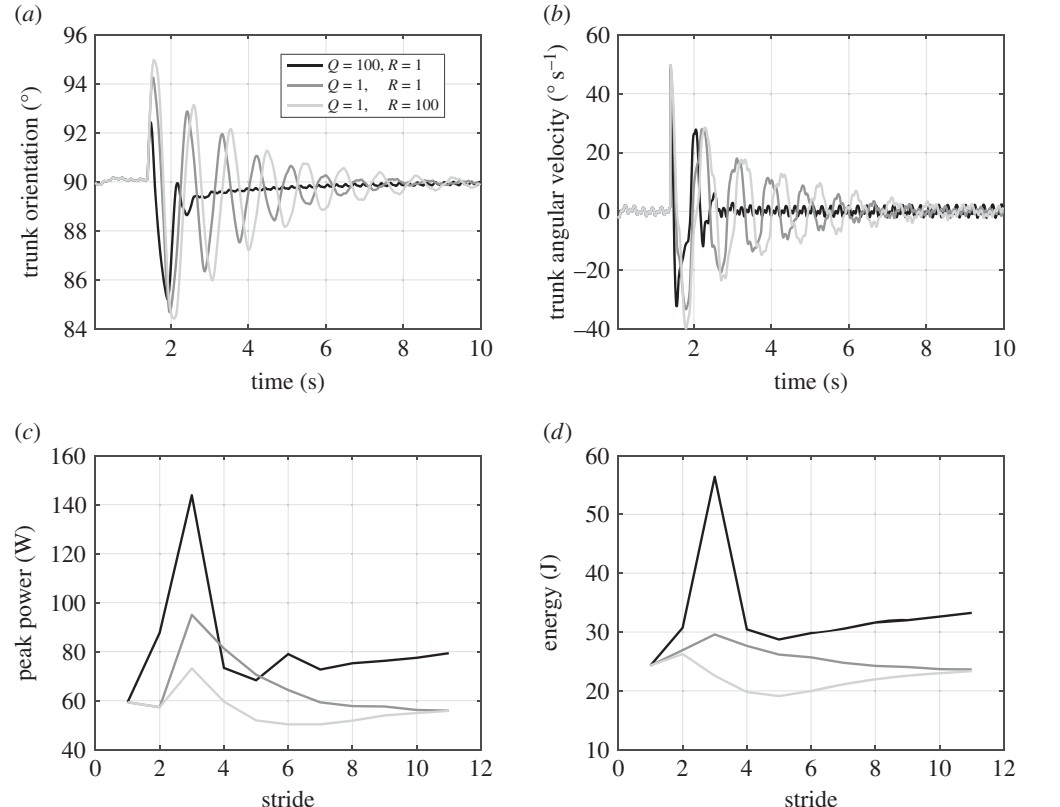

**Figure 7.** Variations of the adaptive nmF model to weighting matrices $Q$ and $R$ in the LQR controller. (a,b) Trunk orientation and its angular velocity for different values of $Q$ and $R$, (c,d) peak power and consumed energy during each stride at the hip joint for different values of $Q$ and $R$.

comparison, the basin of attraction—identified by a range of initial trunk angles and angular velocities in which a model can predict stable walking—is used. For computing the basin of attraction, we perturbed both the trunk angle (from 60 to 120° with a resolution of 1°) and its velocity (from −200 to 150° s$^{-1}$ with a resolution of 0.1 rad s$^{-1}$). Then, those sets of initial conditions in the space $(\phi, \dot{\phi})$ for which models can maintain stable walking are considered as the basin of attraction for that model. The results displayed in figure 8 reveal that the size of basin of attraction in the nmF model is approximately twice that of the FMCH model. Also, this figure shows that the adaptive nmF model has the largest basin of attraction, as expected. Expansion of the basin of attraction from top-left to bottom-right is clearly observable using neuromuscular model and high-level gain adaptation. These results show that our bioinspired neuromuscular models (e.g. nmF) are considerably less sensitive to the initial conditions (or probable perturbations) compared to the biomechanical models (e.g. FMCH).

# 4. Discussion

There is a long debate about the applicability of conceptual models [1,3] on legged locomotion [2]. It is still challenging to translate biological movements into robotic systems. In order to achieve this goal, it is important to learn from biology without getting stuck in details of modelling the system's design and control. For this, mechanical template models [1] can help better understand the basic principles of legged locomotion. These models can then be extended to anchor models such that they can be implemented on robots or be verified in biological gait experiments. One of such template models is based on the virtual pivot point (VPP) concept [11], which is selected for posture control in this paper. Previously, we used the FMCH model [12] as a representation of the VPP concept. However, on a structural level, this control model does not describe how such kind of feedback-augmented passive template models can be realized in the human body by neuromuscular control. Here, we demonstrated the functional coherence of force modulated compliant hip (FMCH) and positive leg force feedback on hip muscles (nmF). With this, we can transfer the (mechanical) template control

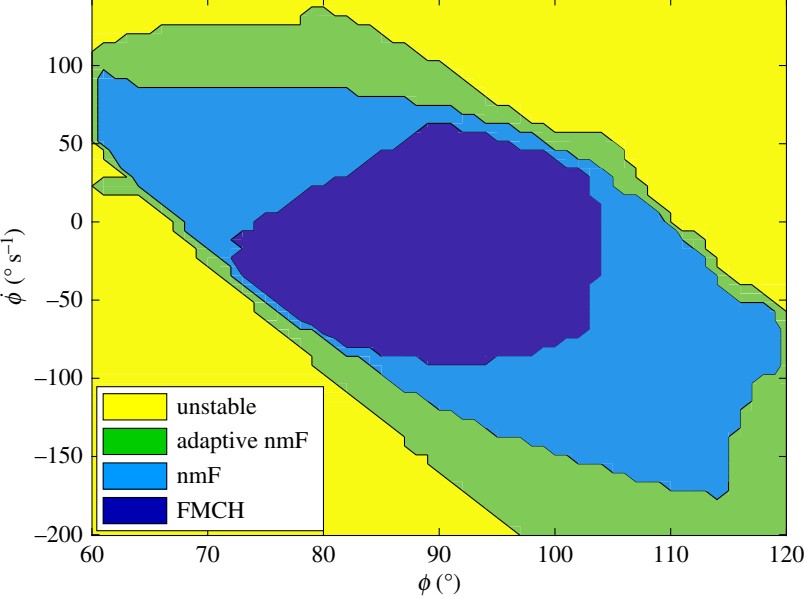

**Figure 8.** Basins of attraction for FMCH, nmF and adaptive nmF models.

structure directly to the neuromuscular control structure. This highlights that in this context, the muscle function can be simplified to an adjustable mechanical spring sharing the same control approach (positive leg force feedback) for postural balance. With this, a descriptive theory of FMCH can be translated to a synthetic theory of designing neuromuscular models for biological movements.

In addition to the muscles' intrinsic properties as part of the biological actuation system, their interplay in the musculoskeletal system can be beneficial for motor control. The contribution of biarticular muscles with well-defined parameters (e.g. lever arm ratio) in control of GRF direction was shown in human perturbed standing [35] and walking experiments [36]. Using GRF direction control with biarticular muscles, the VPP can be obtained by adjustment of muscle stiffness based on the leg force feedback [35]. Here, it was shown that considering 2:1 as the ratio of hip to knee lever arms of biarticular thigh muscles (modelled by springs), the segmented leg model can be simplified into the FMCH model [35]. Hence, in our model, the hip muscles are mimicking RF and HAM muscles in the human body. To validate these theoretical findings for robotic applications, we previously demonstrated that biarticular muscles outperform monoarticular ones in control of GRF direction by comparing them in the humanoid legged robot BioBiped3 [24]. These achievements show how the human musculoskeletal body design facilitates posture control and how it can be easily implemented on a biped robot.

From the neuro-mechanical point of view, our proposed neuromuscular model predicts the emergence of the VPP. To demonstrate that, the GRFs were plotted with respect to the coordinate system centred at body CoM and aligned with trunk orientation in figure 9 for FMCH, nmF and adaptive nmF models. In this figure, the estimated locations of VPP and CoM are also displayed with red and green circles, respectively. The existence of an emerging intersection point shows that the addition of muscle properties (e.g. damping effect represented by force–velocity relation) does not result in deviating from the previous FMCH balance control concept. Moreover, it supports that human muscles are able to control the upper body in a way that the VP concept holds. Hence, this phenomenon is not just an observation, but a control principle.

Three different levels of design and control were addressed in this paper: (i) biomechanical design (ii) neuromuscular control (inner loop) and (iii) event-based control (outer loop). The nmF and adaptive nmF models were developed to focus on the second and third mentioned levels. Higher performance and robustness could be achieved by tuning of the muscle mechanical properties and the control parameters (reflex gains). This shows that in addition to the reflex control based on the sensor-motor maps [37], higher level responses to perturbations can considerably enhance the locomotion robustness. As shown in figure 8, the evolution of the models results in considerable enlargement of the basin of attraction, meaning increase in robustness against postural perturbations. A similar control hierarchy was observed in perturbation recovery when a small step inferred via visual scene is reflected in a decreasing erector spinae stimulation and a forward trunk rotation [38]. Furthermore, it

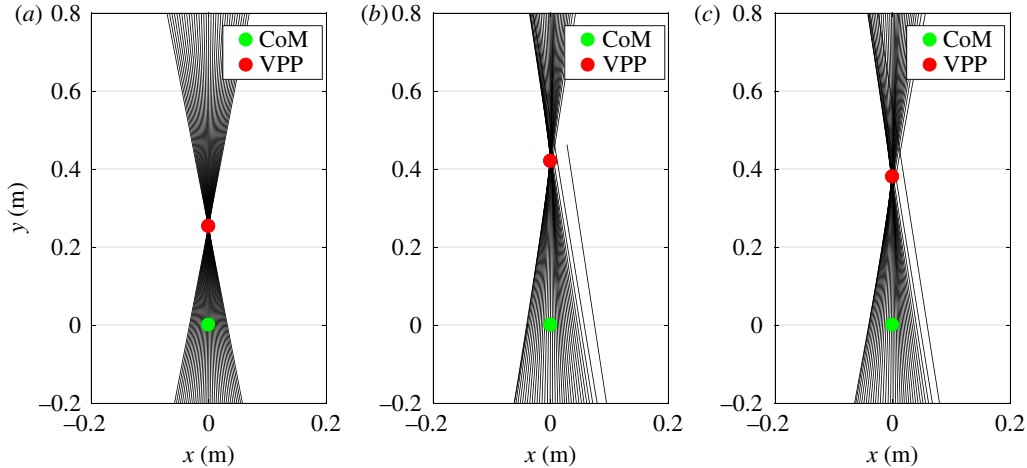

**Figure 9.** Ground reaction forces plotted with respect to trunk coordinate system centred at CoM for FMCH, nmF and adaptive nmF models. The red and green circles illustrate the VPP and CoM location, respectively. As can be seen from these figures, the concept of VPP is also realized with the model presented in this paper. (*a*) FMCH, (*b*) nmF and (*c*) adaptive nmF.

was shown that the motor cortex (MCx) issues voluntary motor commands and mediates reflex-like responses to stretch upper limb muscles [39]. Damaged MCx or corticospinal tract results in severe walking deficits in humans, e.g. foot drop [40,41]. Therefore, high-level event-based control of the reflex gains in the proposed posture control is also supported by evidence in biological locomotor systems. Our proposed LQR controller enables a compromise between efficiency and robustness. Figure 7 shows how a trade-off between $R$ and $Q$ could be made based on the importance of energy efficiency and tracking performance or robustness against perturbations. Such a parameter setting in the higher level control might reflect human-specific behaviours and habits in daily life. Although the cost function in LQR includes consumed energy it can also address peak power, as can be seen in figure 7*c*.

In addition to the aforementioned findings, we have previously employed the nmF model to describe gait asymmetries in stroke patients.[3] This model predicted that the observed gait asymmetry was not caused by deficits in the balance control, but was rather due to changes in stance and swing leg control which is in line with findings in [42]. Such insights can help design function-specific approaches to assist human locomotion, e.g. using exoskeletons. In some studies on neuromuscular system behaviour after stroke, the muscle properties (volume [42], activation [43]) were analysed, while in others the effects on locomotor subfunctions were investigated [44,45]. However, the huge number of parameters in a detailed complex walking model makes understanding of the general concepts of impaired gaits quite difficult. These concepts become more visible in a more abstract level using the proposed template-based biomechanical models. For example, in a detailed model developed in OpenSim (https://simtk.org/projects/hemigait), 23 d.f. and 54 muscle actuators are considered and each muscle comprises many different parameters. Extracting high-level concepts from these numerous parameters is almost impossible unless there exists an abstract model to explain variations in walking and not in each muscle property. As there is no similar architecture (e.g. muscle-like actuators) in exoskeletons, understanding the underlying mechanisms for asymmetry in stroke walking based on the locomotor subfunction concept can help design and control assistive devices. In [46], we showed that implementation of the FMCH concept on a lower-extremity powered exoskeleton (LOPES II) supports this control approach for assisting (reducing metabolic cost) healthy subjects. In future, similar methods can be implemented on assistive devices for rehabilitation training with patients.

In summary, the key advantages of our newly developed neuromuscular model can be categorized as follows: (i) presenting a neuromechanical model to start the journey from template to anchors, (ii) presenting a proof of concept to show that by considering physiological body properties (muscle mechanics and neural control) FMCH can be translated into the human locomotor system, (iii) demonstrating the stiffness modulation as described in FMCH in the neuromuscular system, (iv) introducing LQR as an adaptation method for higher control level which can increase robustness and

---

[3]The results are not described in this paper, but reported in 'Report on neuro-mechanical modelling of neurologically impaired gait' (Deliverable 7.4) which is publicly available in http://www.balance-fp7.eu/private_area/archivo.php?archivo=34.

efficiency. The modulation of reflex gain is also observed in human locomotion and can be represented in future models by reflex networks with higher topological levels (e.g. neural networks modulating reflex gains).

Our future work involves establishing a more elaborate model for human locomotion as in an extended nmF model with segmented legs. In order to develop assistive devices with higher quality inspired by biological gaits, the concepts need to be implemented on more detailed models and robots. These extended models will help to get a better match to gait kinematics and kinetics, e.g. with respect to contact time and push-off mechanics. Our model predicted larger duty factors (stance period over gait cycle duration) than what was observed experimentally, due to the inheritance of the VPP concept [11]. For instance, a matching FMCH-based control could also be present at the ankle joint. In fact, the concerted action of different leg joints and even of the upper body could together contribute to the postural balance which can be represented by the VPP [47]. Such an extension of the model could help to better understand the interplay between ankle and hip strategies for balance control in both normal and pathological gaits.

# 5. Conclusion

In this study, balance control in a neuromuscular bipedal TSLIP model, which represents a template for human walking, is achieved through activating hip muscles (RF and HAM) proportional to the leg force feedback. We demonstrated that this positive feedback of leg force as muscle activation signals is sufficient for ensuring a stable walking gait and also supports the VP concept that is observed in humans. Moreover, we showed that the model by itself is able to increase the range of tolerable disturbances and convergence speed to steady-state walking motion after perturbation. This is attributed to the mechanical function of hip muscles. To further enlarge the domain of attraction and consequently increase robustness, we used a discrete LQR controller designed using the Poincaré map. These results can be interpreted as the outer-loop control. Such a higher level of control can also be found in the neural system like corticospinal layer. This aligns with findings in human walking experiments [38–41].

Both BTSLIP and FMCH are mechanical conceptual models which fail to describe the neuromuscular structure of the human body. In order to validate the value of the predictions made by these models, we need to test them in a more human-like body structure. The nmF model is an attempt to overcome this limitation by representing a pair of thigh muscles and its neural control. In our study, the primary outcome is not to show any advantages of the nmF model compared to the other two conceptual models. Instead, we prove that the concepts hold for a more human-like structure of the model (the idea of an anchor in relation to a template, [1]). With the nmF model, we can now investigate in more detail which structural and functional conditions (e.g. muscle properties and arrangements) are required for a given motor task (e.g. walking). This study is a step towards anchoring a conceptual model. The FMCH model approach with adjustable compliance is a mechanically plausible implementation of the bioinspired VPP concept. However, it is not a biologically plausible realization of the posture control concept. Here, we tried to show that by considering muscle properties, the concept is also biologically plausible and can result in control enhancement. In addition, the proposed adaptive nmF could be considered as a new method to model a higher level of control, e.g. from the spinal cord or the brain in humans. Such an adaptation increases the robustness of the gait against perturbations. This could also open a new door in developing hierarchical neural controls, e.g. by implementing reflex-based and central pattern generators at different control layers.

Data accessibility. All simulations are performed in Matlab/Simulink. The codes are available in a folder in the electronic supplementary material. Human experimental data for moderate walking speed collected from [34] are also presented in the same folder. There is also a file named *main.m* in the same directory that reproduces the figures presented in this paper.

Authors' contributions. A.D. developed the simulation models, implemented control approaches, prepared the results and contributed in data analysis. O.M. contributed in data analysis, writing the manuscript and providing the descriptive figures (figures 1 and 2). M.A.S. is the corresponding author of the article, responsible for the conception and design of simulations and analysis and interpretation of data and writing of the manuscript; A.S. contributed in conception of the study, to discussions and commenting on and writing the article. All authors gave final approval for participation.

Competing interests. We declare we have no competing interests.
Funding. This research was partially supported by INSF under grant no. 95849456 and partially by the German Research Foundation (DFG) under grants nos. AH307/2-1 and SE1042/29-1.
Acknowledgements. We thank colleagues in the BALANCE (http://www.balance-fp7.eu/) project who contributed in discussions about this research and advised on modelling studies.

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
