## [Reviewer comments · Royal Society Open Science]

Review History

RSOS-180380.R0 (Original submission)

Review form: Reviewer 1

Is the manuscript scientifically sound in its present form?

Yes

Are the interpretations and conclusions justified by the results?

No

Is the language acceptable?

Yes

Is it clear how to access all supporting data?

Yes

Do you have any ethical concerns with this paper?

No

Have you any concerns about statistical analyses in this paper?

No

Recommendation?

Major revision is needed (please make suggestions in comments)

Comments to the Author(s)

Dear authors of "Leg Force Guides Posture Control in Human Walking"

I appreciate the technical soundness of the methods in this paper and the detailed provision of methods, data and codes. However, I am not convinced of some central claims in the abstract, title, discussion and conclusion sections. And I believe they are not substantiated by the results, as they stand. For instance, it is not clear that the addition of muscles to the previous iteration of this model (FMCH) has substantially improved its usefulness. So, I suggest further calculations to make the usefulness of the muscle-based modifications to the FMCH model clearer. Also, I think the clarity, flow, grammar and punctuation of the manuscript can be improved through more careful reading by the authors.

More detailed comments and suggestions are provided below:

Scientific comments and suggestions:

- The title of the paper suggests that the present work has discovered how postural control takes place in real human walking. I take note that the use of leg-force feedback in this paper generated human-like virtual pendulum features in simulation. However, there is not enough evidence in this paper to claim whether such leg-force feedback is merely necessary or both necessary and sufficient for real-life postural control during walking. Moreover, the ability to make such a declaration seems to be outside the scope of the purely conceptual and simulation-based methods outlined here. I suggest another title along the lines of "A leg-force feedback controller generates stable biped walking" so that there is no implication about causality or insight into real human walking control based on a purely conceptual model.
- A central tenet of this paper is that the addition of muscles at the hip to the FMCH model is better than having springs (compared to the previous iteration of this model, cited in reference 11). However, I am not convinced that the addition of muscles in the present model (nmF) has improved its previous version in any meaningful way. The only place where a direct comparison is made to the previous FMCH model is in the table with the range of perturbations (table 3). However, just the range alone is not enough to compare the stability. It is important to see the areas of the whole basins of attraction superposed, to be able to judge which model is truly more stable. Similarly, I would like to see comparisons made between the FMCH and nmF models in terms of speed of recovery from perturbations and other such measures (Figures 4 and 6 do not compare FMCH to nmF).
- The paper briefly acknowledges the absence of position control in the model. However, I think this is a bigger flaw than the paper currently admits to. For example, the swing leg placement in this model depends on hip/CoM velocity alone but, past work has found that human walking foot placement depends on both velocity and position of the hip/CoM. Also, because of the absence of position in the structure of the controller, the authors only explored velocity perturbations to the model. But, in most real-life situations, velocity and position perturbations

go hand-in-hand. Given this, it is unclear if and how much this model can withstand such position perturbations.

Comments and suggestions on writing and figures:

- Clarity and flow of thought in the paper can be improved by further careful reading by the authors.
- Some of the long paragraphs in the introduction about templates does not seem central to the message of the paper. Anchors are spoken about without any explanation as to what they are. Again, not clear if the reader needs to know what an anchor is to follow this paper. Maybe just don't mention these concepts.
- Numerous punctuation and grammatical errors; missing commas, run-on sentences, skipping of 'the' and 'a' when needed, its versus it's etc.
- Inconsistencies in definition and usage of abbreviations. Some abbreviations are not defined enough, like virtual pendulum. Sometimes COM is used and sometimes CoM. And I noticed that CoM was not defined before the first instance of usage.
- In figure 1, the meaning of the different shades of red used and the different shapes of the muscles, if any, is unclear. Again, there are inconsistencies in manner of abbreviating. Font may be a little too small. Other figures and tables are clear.
- Try not to mention results in the introduction. Don't mention future work in the conclusion, that would fit better in the discussion.

Review form: Reviewer 2 (Amy Wu)

Is the manuscript scientifically sound in its present form?

Yes

Are the interpretations and conclusions justified by the results?

Yes

Is the language acceptable?

Yes

Is it clear how to access all supporting data?

Yes

Do you have any ethical concerns with this paper?

No

Have you any concerns about statistical analyses in this paper?

No

Recommendation?

Major revision is needed (please make suggestions in comments)

Comments to the Author(s)

This manuscript is an extension of the previously developed Force Modulated Compliant Hip (FMCH) model. Whereas the previous model used hip springs to actuate the legs, the new model nmF uses Hill-type muscle models of the rectus femoris and hamstring. Discrete LQR was also

used to adapt the reflex parameters during perturbed steps. The model resembles human walking and is an improvement in terms of robustness over the FMCH.

This paper was well-written, and the descriptions of the model were clear. The authors motivated that templates are very useful for explaining complex behavior yet the trunk is still an important component that should not be neglected. The motivation for including hip muscles, however, is a bit unclear – aside from providing a way to anchor the SLIP template. It is not surprising that a more complicated model (nmF) could better capture human behavior than a simpler one (FMCH). What did FMCH or BTSLIP fail to explain? Was the nmF able to explain or capture something that they could not?

The authors stated that their reflex control is a “bioinspired posture control method” in comparison with Geyer’s (ref 23) use of optimization methods. However, LQR inherently is an optimization based on Q and R. Also, Q and R were chosen as identity. How sensitive are the results when giving more importance to error or effort?

In Section 2.3, please explain what is meant by “fast walking is not reachable” since the reflex-based models tend to have problems with slow speeds. If one speed was chosen for comparison, what was the speed chosen?

In comparisons with human data, why is there a time shift in Figure 3? Are model and human walking at the same speed? Toe-off at 80% of gait is quite late. Is there something in the model delaying push-off? Could the authors quantify the difference between model and human trajectories (with cross correlation values perhaps)?

A few items were brought up in Results with little prior context. For example, more explanation is needed for preflex, such as how was the optimal predefined signal derived? And how was power derived? Is that joint power from hip torque and angular velocity? For Table 3, how was recovery defined (e.g. some number of steps taken after the perturbation)?

On page 16, the authors state the main results from their use of the nmF to model the gait of stroke patients. However, the cited paper does not contain any of the co-authors as authors. Do the authors instead mean that the model can predict what the cited paper found?

There seems to be three different main points. In the beginning, the authors motivated the use of simpler models to explain complex human behavior. But then they added a complexity to their model by using muscles. Another main point was that understanding and controlling the trunk is vital to gait behavior. Presumably this is one reason why the authors added hip muscles. Then in the discussion and conclusion, the authors discuss the application of the model on exoskeletons by further increasing the complexity, which seems contradictory to the first point.

I realize that this is an extension and build up of previous work, but the authors themselves are co-authors of over 1/3 of the references cited (mostly Sharbafi). Perhaps Song and Geyer’s 2015 paper on the 3D neuromuscular model should be also included, because his model includes the rectus femoris, reflexes, trunk control, and was tested with push perturbations.

Minor comments:

Section 2.2.1- please briefly include a few of the advantages of VBLA over horizontal velocity.

Section 2.2.3 - please define G^* and U^* .

Section 3 first line - “schems” typo

Fig 3: “weigh” typo, add that the dot signifies toe-off, is Figure 3a x-axis label meant to be gait cycle?

Decision letter (RSOS-180380.R0)

18-May-2018

Dear Dr Sharbafi:

Manuscript ID RSOS-180380 entitled "Leg Force Guides Posture Control In Human Walking" which you submitted to Royal Society Open Science, has been reviewed. The comments from reviewers are included at the bottom of this letter.

In view of the criticisms of the reviewers, the manuscript has been rejected in its current form. However, a new manuscript may be submitted which takes into consideration these comments.

Please note that resubmitting your manuscript does not guarantee eventual acceptance, and that your resubmission will be subject to peer review before a decision is made.

Your resubmitted manuscript should be submitted by 15-Nov-2018. If you are unable to submit by this date please contact the Editorial Office.

Please note that Royal Society Open Science will introduce article processing charges for all new submissions received from 1 January 2018. Charges will also apply to papers transferred to Royal Society Open Science from other Royal Society Publishing journals, as well as papers submitted as part of our collaboration with the Royal Society of Chemistry (<http://rsos.royalsocietypublishing.org/chemistry>). If your manuscript is submitted and accepted for publication after 1 Jan 2018, you will be asked to pay the article processing charge, unless you request a waiver and this is approved by Royal Society Publishing. You can find out more about the charges at <http://rsos.royalsocietypublishing.org/page/charges>. Should you have any queries, please contact openscience@royalsociety.org.

Kind regards,
Thadcha Retneswaran
Royal Society Open Science
openscience@royalsociety.org

on behalf of Dr Monica Daley (Associate Editor) and Prof. R. Kerry Rowe (Subject Editor)
openscience@royalsociety.org

Associate Editor Comments to Author (Dr Monica Daley):

Your paper has been reviewed by two experts, who both praise the technical approach and scientific soundness overall. However, both Reviewers also raise concerns about whether the development of this specific muscle-based model makes a substantive scientific contribution relative to the previously published spring-based FMCH model.

What fundamental insight is gained from the current model that justifies the increased complexity? A more quantitative statistical comparison between models and human experimental data could perhaps help address this.

Additionally, the reviewers have provided some thoughtful specific comments about the presentation and interpretation of the work, which are important for placing the current findings in the broader context of the field.

Considering the comments and recommendations of both reviewers, I cannot accept the paper for publication in RSOS in its current form, but I will consider a substantively revised version that fully addresses the concerns raised.

I hope the authors will carefully consider the comments of both reviewers and consider whether the case for the scientific contribution of the work can be strengthened. The authors should also revise the text to ensure that the conclusions and interpretations are fully justified by the results presented.

Reviewers' Comments to Author:

Reviewer: 1

Comments to the Author(s)

Dear authors of "Leg Force Guides Posture Control in Human Walking"

I appreciate the technical soundness of the methods in this paper and the detailed provision of methods, data and codes. However, I am not convinced of some central claims in the abstract, title, discussion and conclusion sections. And I believe they are not substantiated by the results, as they stand. For instance, it is not clear that the addition of muscles to the previous iteration of this model (FMCH) has substantially improved its usefulness. So, I suggest further calculations to make the usefulness of the muscle-based modifications to the FMCH model clearer. Also, I think the clarity, flow, grammar and punctuation of the manuscript can be improved through more careful reading by the authors.

More detailed comments and suggestions are provided below:

Scientific comments and suggestions:

- The title of the paper suggests that the present work has discovered how postural control takes place in real human walking. I take note that the use of leg-force feedback in this paper generated human-like virtual pendulum features in simulation. However, there is not enough evidence in this paper to claim whether such leg-force feedback is merely necessary or both necessary and sufficient for real-life postural control during walking. Moreover, the ability to make such a declaration seems to be outside the scope of the purely conceptual and simulation-based methods outlined here. I suggest another title along the lines of "A leg-force feedback controller generates stable biped walking" so that there is no implication about causality or insight into real human walking control based on a purely conceptual model.

- A central tenet of this paper is that the addition of muscles at the hip to the FMCH model is better than having springs (compared to the previous iteration of this model, cited in reference 11). However, I am not convinced that the addition of muscles in the present model (nmF) has improved its previous version in any meaningful way. The only place where a direct comparison is made to the previous FMCH model is in the table with the range of perturbations (table 3). However, just the range alone is not enough to compare the stability. It is important to see the

areas of the whole basins of attraction superposed, to be able to judge which model is truly more stable. Similarly, I would like to see comparisons made between the FMCH and nmF models in terms of speed of recovery from perturbations and other such measures (Figures 4 and 6 do not compare FMCH to nmF).

- The paper briefly acknowledges the absence of position control in the model. However, I think this is a bigger flaw than the paper currently admits to. For example, the swing leg placement in this model depends on hip/CoM velocity alone but, past work has found that human walking foot placement depends on both velocity and position of the hip/CoM. Also, because of the absence of position in the structure of the controller, the authors only explored velocity perturbations to the model. But, in most real-life situations, velocity and position perturbations go hand-in-hand. Given this, it is unclear if and how much this model can withstand such position perturbations.

Comments and suggestions on writing and figures:

- Clarity and flow of thought in the paper can be improved by further careful reading by the authors.
- Some of the long paragraphs in the introduction about templates does not seem central to the message of the paper. Anchors are spoken about without any explanation as to what they are. Again, not clear if the reader needs to know what an anchor is to follow this paper. Maybe just don't mention these concepts.
- Numerous punctuation and grammatical errors; missing commas, run-on sentences, skipping of 'the' and 'a' when needed, its versus it's etc.
- Inconsistencies in definition and usage of abbreviations. Some abbreviations are not defined enough, like virtual pendulum. Sometimes COM is used and sometimes CoM. And I noticed that CoM was not defined before the first instance of usage.
- In figure 1, the meaning of the different shades of red used and the different shapes of the muscles, if any, is unclear. Again, there are inconsistencies in manner of abbreviating. Font may be a little too small. Other figures and tables are clear.
- Try not to mention results in the introduction. Don't mention future work in the conclusion, that would fit better in the discussion.

Reviewer: 2

Comments to the Author(s)

This manuscript is an extension of the previously developed Force Modulated Compliant Hip (FMCH) model. Whereas the previous model used hip springs to actuate the legs, the new model nmF uses Hill-type muscle models of the rectus femoris and hamstring. Discrete LQR was also used to adapt the reflex parameters during perturbed steps. The model resembles human walking and is an improvement in terms of robustness over the FMCH.

This paper was well-written, and the descriptions of the model were clear. The authors motivated that templates are very useful for explaining complex behavior yet the trunk is still an important component that should not be neglected. The motivation for including hip muscles, however, is a bit unclear – aside from providing a way to anchor the SLIP template. It is not surprising that a more complicated model (nmF) could better capture human behavior than a simpler one (FMCH). What did FMCH or BTSLIP fail to explain? Was the nmF able to explain or capture something that they could not?

The authors stated that their reflex control is a “bioinspired posture control method” in comparison with Geyer's (ref 23) use of optimization methods. However, LQR inherently is an

optimization based on Q and R. Also, Q and R were chosen as identity. How sensitive are the results when giving more importance to error or effort?

In Section 2.3, please explain what is meant by “fast walking is not reachable” since the reflex-based models tend to have problems with slow speeds. If one speed was chosen for comparison, what was the speed chosen?

In comparisons with human data, why is there a time shift in Figure 3? Are model and human walking at the same speed? Toe-off at 80% of gait is quite late. Is there something in the model delaying push-off? Could the authors quantify the difference between model and human trajectories (with cross correlation values perhaps)?

A few items were brought up in Results with little prior context. For example, more explanation is needed for preflex, such as how was the optimal predefined signal derived? And how was power derived? Is that joint power from hip torque and angular velocity? For Table 3, how was recovery defined (e.g. some number of steps taken after the perturbation)?

On page 16, the authors state the main results from their use of the nmF to model the gait of stroke patients. However, the cited paper does not contain any of the co-authors as authors. Do the authors instead mean that the model can predict what the cited paper found?

There seems to be three different main points. In the beginning, the authors motivated the use of simpler models to explain complex human behavior. But then they added a complexity to their model by using muscles. Another main point was that understanding and controlling the trunk is vital to gait behavior. Presumably this is one reason why the authors added hip muscles. Then in the discussion and conclusion, the authors discuss the application of the model on exoskeletons by further increasing the complexity, which seems contradictory to the first point.

I realize that this is an extension and build up of previous work, but the authors themselves are co-authors of over 1/3 of the references cited (mostly Sharbafi). Perhaps Song and Geyer’s 2015 paper on the 3D neuromuscular model should be also included, because his model includes the rectus femoris, reflexes, trunk control, and was tested with push perturbations.

Minor comments:

Section 2.2.1- please briefly include a few of the advantages of VBLA over horizontal velocity.

Section 2.2.3 – please define G^* and U^* .

Section 3 first line – “schems” typo

Fig 3: “weighth” typo, add that the dot signifies toe-off, is Figure 3a x-axis label meant to be gait cycle?

Author's Response to Decision Letter for (RSOS-180380.R0)

See Appendix A.

RSOS-181911.R0

Review form: Reviewer 1

Is the manuscript scientifically sound in its present form?

Yes

Are the interpretations and conclusions justified by the results?

Yes

Is the language acceptable?

Yes

Is it clear how to access all supporting data?

Yes

Do you have any ethical concerns with this paper?

No

Have you any concerns about statistical analyses in this paper?

No

Recommendation?

Accept with minor revision (please list in comments)

Comments to the Author(s)

See attached file (Appendix B).

Review form: Reviewer 2

Is the manuscript scientifically sound in its present form?

Yes

Are the interpretations and conclusions justified by the results?

Yes

Is the language acceptable?

Yes

Is it clear how to access all supporting data?

Yes

Do you have any ethical concerns with this paper?

No

Have you any concerns about statistical analyses in this paper?

No

Recommendation?

Major revision is needed (please make suggestions in comments)

Comments to the Author(s)

The purpose of this manuscript is to extend a previously developed Force Modulated Compliant Hip (FMCH) "template" model towards a neuromuscular "anchor" model. Whereas the previous model used hip springs to actuate the legs, the new model nmF uses Hill-type muscle models of the rectus femoris and hamstring. Discrete LQR was also used to adapt the reflex parameters during perturbed steps. The model resembles human walking and is an improvement in terms of robustness over the FMCH.

The revised manuscript is a great improvement, and the authors have addressed my major concerns. The motivation of the manuscript is much clearer now (template to anchor models), in the text and with the addition of Figure 1.

A few remaining comments:

Page 10 Line 22 to 41 - This is a confusing paragraph. Since there are two hip muscles in the nMF model, it is unclear how the vastus was included or used. The text also jumps between the current model and other models, which makes it more confusing. Perhaps it would be better to put the details about the authors' model first and then note how it is different from others.

Old references seem to have been submitted in error. There are several citations with question marks. In the response to reviewers, the authors stated they have modified the references, but the references submitted are the same as before.

Minor comments:

Page 2 Line 50 - It is unclear what is meant by engineering techniques. Can the authors provide examples from the cited works?

Page 5 Line 32 - what does "in the sequel" refer to?

Page 8 Line 14 - "at THE hip joint"

Page 9 Line 12 - is 50 steps the authors' definition or a known standard? It is worded as if it is the standard. Perhaps "... if it can keep stability, specified here as 50 steps after perturbation occurrence" would be clearer. I also suggest using "maintain," instead of "keep."

Page 9 Line 20, Page 10 Line 49 - do not use contractions ("can't")

Page 11 Line 9 - Perhaps "we compare" is more explicit than "we examine" because the reflex control is later compared against adaptive nmF.

Page 12 Line 53 - Note the achievable range, instead of just the upper limit (as the authors have clarified in the response)

Page 13 Line 50 - Was stride time the same between model and human?

Page 14 Eqn 25 - I do not think τ_1 and τ_2 were already defined

Page 14 Line 49 - Include that both energy and peak power are calculated. Otherwise it is confusing why the equation for work is shown, but the results are in terms of power.

Page 1 - Merely to further assist the reader: adding labels (BSLIP, BTSLIP, FMCH, nmF) onto the A-D figures

Decision letter (RSOS-181911.R0)

23-Jan-2019

Dear Dr Sharbafi

On behalf of the Editor, I am pleased to inform you that your Manuscript RSOS-181911 entitled "Positive Leg Force Feedback on Hip Muscles Supports Postural Stability in Bipedal Walking" has been accepted for publication in Royal Society Open Science subject to minor revision in accordance with the referee suggestions. Please find the referees' comments at the end of this email.

The reviewers and Subject Editor have recommended publication, but also suggest some minor revisions to your manuscript. Therefore, I invite you to respond to the comments and revise your manuscript.

- Ethics statement

- Data accessibility

If you wish to submit your supporting data or code to Dryad (<http://datadryad.org/>), or modify your current submission to dryad, please use the following link:
<http://datadryad.org/submit?journalID=RSOS&manu=RSOS-181911>

- Competing interests

- Authors' contributions

AB carried out the molecular lab work, participated in data analysis, carried out sequence alignments, participated in the design of the study and drafted the manuscript; CD carried out

the statistical analyses; EF collected field data; GH conceived of the study, designed the study, coordinated the study and helped draft the manuscript. All authors gave final approval for publication.

- Acknowledgements

- Funding statement

Because the schedule for publication is very tight, it is a condition of publication that you submit the revised version of your manuscript before 01-Feb-2019. Please note that the revision deadline will expire at 00.00am on this date. If you do not think you will be able to meet this date please let me know immediately.

Supplementary files will be published alongside the paper on the journal website and posted on the online figshare repository (<https://figshare.com>). The heading and legend provided for each supplementary file during the submission process will be used to create the figshare page, so

please ensure these are accurate and informative so that your files can be found in searches. Files on figshare will be made available approximately one week before the accompanying article so that the supplementary material can be attributed a unique DOI.

on behalf of Dr Monica Daley (Associate Editor) and Professor R. Kerry Rowe (Subject Editor)
openscience@royalsociety.org

Associate Editor Comments to Author (Dr Monica Daley):

Thank you for your patience in waiting for a decision on this manuscript. We have now received feedback from the referees, who are happy with the revised version of the paper, and suggest only a few minor additional changes to the text. I am therefore happy to accept the paper for publication, subject to addressing these final comments. Please ensure that the reference list is correctly updated, as noted by Reviewer 2.

Reviewer comments to Author:
Reviewer: 2

Comments to the Author(s)

The purpose of this manuscript is to extend a previously developed Force Modulated Compliant Hip (FMCH) "template" model towards a neuromuscular "anchor" model. Whereas the previous model used hip springs to actuate the legs, the new model nmF uses Hill-type muscle models of the rectus femoris and hamstring. Discrete LQR was also used to adapt the reflex parameters during perturbed steps. The model resembles human walking and is an improvement in terms of robustness over the FMCH.

The revised manuscript is a great improvement, and the authors have addressed my major concerns. The motivation of the manuscript is much clearer now (template to anchor models), in the text and with the addition of Figure 1.

A few remaining comments:

Page 10 Line 22 to 41 - This is a confusing paragraph. Since there are two hip muscles in the nmF model, it is unclear how the vastus was included or used. The text also jumps between the current model and other models, which makes it more confusing. Perhaps it would be better to put the details about the authors' model first and then note how it is different from others.

Old references seem to have been submitted in error. There are several citations with question marks. In the response to reviewers, the authors stated they have modified the references, but the references submitted are the same as before.

Minor comments:

Page 2 Line 50 - It is unclear what is meant by engineering techniques. Can the authors provide examples from the cited works?

Page 5 Line 32 - what does "in the sequel" refer to?

Page 8 Line 14 – “at THE hip joint”

Page 9 Line 12 – is 50 steps the authors’ definition or a known standard? It is worded as if it is the standard. Perhaps “... if it can keep stability, specified here as 50 steps after perturbation occurrence” would be clearer. I also suggest using “maintain,” instead of “keep.”

Page 9 Line 20, Page 10 Line 49 – do not use contractions (“can’t”)

Page 11 Line 9 – Perhaps “we compare” is more explicit than “we examine” because the reflex control is later compared against adaptive nMF.

Page 12 Line 53 – Note the achievable range, instead of just the upper limit (as the authors have clarified in the response)

Page 13 Line 50 – Was stride time the same between model and human?

Page 14 Eqn 25 – I do not think τ_1 and τ_2 were already defined

Page 14 Line 49 – Include that both energy and peak power are calculated. Otherwise it is confusing why the equation for work is shown, but the results are in terms of power.

Page 1 – Merely to further assist the reader: adding labels (BSLIP, BTSLIP, FMCH, nMF) onto the A-D figures

Reviewer: 1

Comments to the Author(s)

See attached file

Author's Response to Decision Letter for (RSOS-181911.R0)

See Appendix C.

Decision letter (RSOS-181911.R1)

08-Feb-2019

Dear Dr Sharbafi,

I am pleased to inform you that your manuscript entitled "From Template to Anchors: Transfer of VPP balance template to adaptive neuromuscular gait model increases walking stability" is now accepted for publication in Royal Society Open Science.

on behalf of Dr Monica Daley (Associate Editor) and Professor R. Kerry Rowe (Subject Editor)
openscience@royalsociety.org

**Authors' response to the reviewers' comments on the
Paper No. RSOS-180380**

Leg Force Guides Posture Control In Human Walking

Dear Dr. *Daley*,

We would like to thank you and the reviewers for the constructive and helpful comments and suggestions. We considered all of them and revised the paper accordingly. To address the comment, suggestions, and concerns, we modified the paper and added new explanations/clarifications.

Editor's comment

Associate Editor Comments to Author (Dr Monica Daley):

Your paper has been reviewed by two experts, who both praise the technical approach and scientific soundness overall. However, both Reviewers also raise concerns about whether the development of this specific muscle-based model makes a substantive scientific contribution relative to the previously published spring-based FMCH model.

What fundamental insight is gained from the current model that justifies the increased complexity? A more quantitative statistical comparison between models and human experimental data could perhaps help address this.

Additionally, the reviewers have provided some thoughtful specific comments about the presentation and interpretation of the work, which are important for placing the current findings in the broader context of the field.

Considering the comments and recommendations of both reviewers, I cannot accept the paper for publication in RSOS in its current form, but I will consider a substantively revised version that fully addresses the concerns raised.

I hope the authors will carefully consider the comments of both reviewers and consider whether the case for the scientific contribution of the work can be strengthened. The authors should also revise

the text to ensure that the conclusions and interpretations are fully justified by the results presented.

We thank the editors and reviewers for their time and critical comments, which have helped us improve the manuscript. We have made changes accordingly. In order to address all comments and suggestions raised by the reviewers and the associate editor, we have fully revised the paper to improve clarification, emphasize on the main contributions of the study and correct writing errors. All the changes are shown by blue color in this letter and in the revised manuscript.

The main modifications can be summarized as:

1. The main motivation of extending the FMCH model to nmF is clarified and supported by new results (see 4). The fundamental finding with the nmF model is its ability (as a neuromuscular and not just mechanical mode) to replicate human balance control based on FMCH concept and this is now highlighted in the discussion and conclusion sections.
2. A new figure (Fig. 1) is added to explain the evolution of modeling from template to anchor level.
3. New quantitative comparisons between human and model regarding FMCH and nmF models are presented (Results section) and discussed (Discussion section).
4. Comparison between FMCH, nmF and adaptive nmF is extended by adding new figures or modifying the previous ones:
 - a. Figures 3, 4 and 6 in the previous version are adapted by adding the results of FMCH and increasing the perturbation magnitude (Figures 4, 5 and 6 in the revised version).
 - b. Fig. 5 of the previous version is revised by analyzing the VPP point with FMCH and adaptive nmF models (Fig. 9 in the revised version).
 - c. A new figure (Fig. 7 in the revised version) is added to demonstrate the effects of tuning R and Q parameters in LQR controller for balancing trade off between performance and efficiency
 - d. A new figure (Fig. 8 in the revised version) is added to compare the basin of attraction of the three methods as a measure of robustness against perturbation
5. We have also investigated response to ground level perturbation which is not reported in the manuscript, but here in the response letter (see below).
6. The discussion section is extended based on the new aforementioned results.

In the following, a detailed response to the comments is given. The comments and our replies are in ***bold italic*** and normal faces, respectively.

Reviewer 1:

Dear authors of "Leg Force Guides Posture Control in Human Walking" □□I appreciate the technical soundness of the methods in this paper and the detailed provision of methods, data and codes. However, I am not convinced of some central claims in the abstract, title, discussion and conclusion sections. And I believe they are not substantiated by the results, as they stand. For instance, it is not clear that the addition of muscles to the previous iteration of this model (FMCH) has substantially improved its usefulness. So, I suggest further calculations to make the usefulness of the muscle-based modifications to the FMCH model clearer. Also, I think the clarity, flow, grammar and punctuation of the manuscript can be improved through more careful reading by the authors. □□More detailed comments and suggestions are provided below:

- ✓ We are thankful for the very precise and solid comments of the reviewer. We agree that the discussion in the previous version was not broad enough and we improved it accordingly.
- ✓ Here, we now clarify the goal of extending the model from a biomechanical to neuromuscular level. We revised the manuscript to be more concrete and understandable. In summary:
 - o It is important to say that the first goal of this extension was not achieving higher performance (even if it is obtained consequently). The target was verifying if the FMCH concept which was based on the behavioral results in human locomotion is explainable with the neuromuscular properties of the human body. It is clear that we do not have adjustable springs in the body, but muscles which are controlled by neural activation. Therefore, considering the muscle properties and the neural control, we aimed at investigating whether these ingredients can result in FMCH and VPP.
 - o So far it was not clear how mechanical template models relate to more physiological neuromuscular gait models. Here we demonstrate the functional coherence of force modulated compliant hip (FMCH) and positive leg force feedback on hip muscles (nmF). With this we can transfer the (mechanical) template control structure directly to the neuromuscular control structure. This highlights that in this context, the muscle function can be simplified to an adjustable mechanical spring sharing the same control approach (positive leg force feedback) for postural balance. With this a descriptive theory of FMCH can be translated to a synthetic theory of designing neuromuscular models for biological movements.
 - o By making the model more complex, other benefits such as improved performance, robustness and efficiency can be achieved as well. As the second goal of this study, we present now more evidences about the advantages of nmF and adaptive nmF compared

to FMCH. For this, we executed a number of optimizations to find the best performance of each method for fair comparison.

- ✓ We have revised the discussion and conclusion benefitting from your comments. It is now more structured, covers other works and is broader as well.
- ✓ The whole paper was carefully revised by the authors to improve flowness and to remove typos and grammatical errors .

□□ *Scientific comments and suggestions:* □-

The title of the paper suggests that the present work has discovered how postural control takes place in real human walking. I take note that the use of leg-force feedback in this paper generated human-like virtual pendulum features in simulation. However, there is not enough evidence in this paper to claim whether such leg-force feedback is merely necessary or both necessary and sufficient for real-life postural control during walking. Moreover, the ability to make such a declaration seems to be outside the scope of the purely conceptual and simulation-based methods outlined here. I suggest another title along the lines of "A leg-force feedback controller generates stable biped walking" so that there is no implication about causality or insight into real human walking control based on a purely conceptual model. □□

- ✓ We agree that proving causality and control technique in human body is not possible by computer simulation model results even if they would be identical to findings in human experiment. In this study, we show for the first time that positive leg force feedback on hip muscles is predicted to support posture control in bipedal walking. Additionally, our results suggest that a force modulated hip compliance can be transferred to a force modulated hip muscle activation, both providing postural stability in walking.
- ✓ We have modified the title based on the suggestion of the reviewer to “positive leg force feedback on hip muscles supports postural stability in bipedal walking”

- A central tenet of this paper is that the addition of muscles at the hip to the FMCH model is better than having springs (compared to the previous iteration of this model, cited in reference 11). However, I am not convinced that the addition of muscles in the present model (nmF) has improved its previous version in any meaningful way. The only place where a direct comparison is made to the previous FMCH model is in the table with the range of perturbations (table 3). However, just the range alone is not enough to compare the stability. It is important to see the areas of the whole basins of attraction superposed, to be able to judge which model is truly more stable. Similarly, I would like to see

comparisons made between the FMCH and nmF models in terms of speed of recovery from perturbations and other such measures (Figures 4 and 6 do not compare FMCH to nmF).

- ✓ Advantages of nmF compared to FMCH are now described in a clearer way in the revised version. We found the following benefits of muscle properties to improve motion performance and robustness:
 - We have added FMCH results to the previous figures (Figs. 3, 4 and 6 in the previous version) to compare the performance of the nmF model (Figs. 4 to 6 in the revised version). The extension of the FMCH model to the neuromuscular level keeps the ability to mimic human behavior while it has significantly higher performance and lower energy consumption.
 - As suggested, we have added a new Figure (Fig. 8 in the revised version) comparing the basin of attraction of the different models. It can be clearly seen that the nmF has a larger basin of attraction compared to FMCH. The adaptive nmF has the largest basin of attraction. These results show a considerable increase in robustness against perturbations: the basin of attraction of the nmF model is twice as large as in FMCH. Therefore, the neuromuscular control can attenuate larger perturbations than the mechanical FMCH model.
 - Table 3 in the previous version is removed. The new Fig. 8 illustrating basin of attraction also includes the data of the omitted table.

□□- *The paper briefly acknowledges the absence of position control in the model. However, I think this is a bigger flaw than the paper currently admits to. For example, the swing leg placement in this model depends on hip/CoM velocity alone but, past work has found that human walking foot placement depends on both velocity and position of the hip/CoM. Also, because of the absence of position in the structure of the controller, the authors only explored velocity perturbations to the model. But, in most real-life situations, velocity and position perturbations go hand-in-hand. Given this, it is unclear if and how much this model can withstand such position perturbations.* □□

- ✓ Our claim is about advantages of using leg force as a reflex signal to tune the muscles contributing to posture control in stance phase when the leg is in contact with the ground. In this condition, control of position and force together is not achievable with such a simple formulation. Instead, we control the stiffness (impedance). Of course we do not blame position control in all cases. As you mentioned, in swing leg adjustment, the direction of the swing leg is adjusted with position control using the VBLA approach. Indeed when the leg is moving freely in

the air, position control is preferred and no need to force control. We have revised the manuscript to prevent such a misinterpretation.

- ✓ In the new results showing basin of attraction, we deviate both trunk angle and angular velocity with certain values to find the stability margin. Therefore, both velocity and position perturbations are considered as suggested.
- ✓ We have also examined ground level perturbations which can be considered as perturbation in position. All methods can equally handle 2cm drops in ground level. For larger values the model switches to running, inherited from the elastic leg property of SLIP model which can not be compensated by posture control. The results for FMCH and BSLIP are shown in Fig. a in the following. As it does not provide additional useful information and given the limited lengths of the paper (we have already 9 figures) we do not present these results in the paper.

Figure a. Response to 2cm ground level perturbations with FMCH. Similar results are found with nmF and Adaptive nmF method.

Comments and suggestions on writing and figures:

- *Clarity and flow of thought in the paper can be improved by further careful reading by the authors.*

- ✓ The paper have been completely revised considering all comments of the reviewers and the associate editor.
- ✓ We have spent more than 3 months on doing new simulations and revising the paper to provide a clear story and consistent writing flow.

- Some of the long paragraphs in the introduction about templates does not seem central to the message of the paper. Anchors are spoken about without any explanation as to what they are. Again,

not clear if the reader needs to know what an anchor is to follow this paper. Maybe just don't mention these concepts. □

- ✓ These concepts are essential for this paper. This holds as template models are required to interpret experimental findings and anchor models are required for technical implementations.
- ✓ Here, we work on filling the gap between template and anchor models.
- ✓ We have shortened the description of template models in the introduction.
- ✓ We have revised the paper to highlight the importance of these two complementary modeling approaches for legged locomotion.
- ✓ We have added a figure (Fig. 1 in the revised version) to show the roadmap of template anchor modeling including the position of our study.

- Numerous punctuation and grammatical errors; missing commas, run-on sentences, skipping of 'the' and 'a' when needed, its versus it's etc.

- ✓ The paper have been revised by the authors to resolve writing issues.

□- *Inconsistencies in definition and usage of abbreviations. Some abbreviations are not defined enough, like virtual pendulum. Sometimes COM is used and sometimes CoM. And I noticed that CoM was not defined before the first instance of usage.*

- ✓ We have added a table (Tab. 1 in the revised version) describing all abbreviations.
- ✓ We have checked the notation thoroughly and make writing style consistent in the whole paper.

□- *In figure 1, the meaning of the different shades of red used and the different shapes of the muscles, if any, is unclear. Again, there are inconsistencies in manner of abbreviating. Font may be a little too small. Other figures and tables are clear.* □

- ✓ We have modified the figure (Fig. 2 in the revised version) by
 - depicting shapes of the muscles, similar to each other so as to prevent any possible misunderstanding,
 - enlarging the fonts to be more readable,
 - adding descriptions to clarify the colors.

- Try not to mention results in the introduction. Don't mention future work in the conclusion, that would fit better in the discussion.

- ✓ Thank you for this helpful suggestion. We have revised the introduction, discussion and conclusion sections following this comment.

Reviewer 2:

Comments to the Author(s)

This manuscript is an extension of the previously developed Force Modulated Compliant Hip (FMCH) model. Whereas the previous model used hip springs to actuate the legs, the new model nmF uses Hill-type muscle models of the rectus femoris and hamstring. Discrete LQR was also used to adapt the reflex parameters during perturbed steps. The model resembles human walking and is an improvement in terms of robustness over the FMCH.

This paper was well-written, and the descriptions of the model were clear. The authors motivated that templates are very useful for explaining complex behavior yet the trunk is still an important component that should not be neglected.

- ✓ Thanks for the nicely summarized description of the work.

The motivation for including hip muscles, however, is a bit unclear—aside from providing a way to anchor the SLIP template. It is not surprising that a more complicated model (nmF) could better capture human behavior than a simpler one (FMCH). What did FMCH or BTSLIP fail to explain? Was the nmF able to explain or capture something that they could not?

- ✓ Both BTSLIP and FMCH are mechanical conceptual models which fail to describe the neuromuscular structure of the human body.
- ✓ In order to prove the value of the predictions made by these models we need to test them in a more human-like body structure. The nmF model is an attempt to overcome this limitation by representing a pair of thigh muscles and its neural control.
- ✓ In our study, the primary outcome is not to show any advantages of the nmF model compared to the two other conceptual models. Instead, we prove that the concepts hold for a more human-like structure of the model (idea of an anchor in relation to a template, Full and Koditschek, 1999). With the nmF model we can now investigate in more detail, which structural and functional conditions (e.g. muscle properties and arrangements) are required for a given motor task (e.g. walking).
- ✓ We agree with you that this paper is a step towards anchoring a conceptual model. This is one important objective of this paper. The FMCH model approach with adjustable compliance is a

mechanically plausible implementation of the bioinspired VPP concept. However, it is not yet biologically plausible realization of the posture control concept. Here, we try to show that considering muscle properties the concept is also biologically plausible and can result in control enhancement.

- ✓ In addition, the proposed adaptive nmF could be considered as a new method to model a higher level of control which was missing in the FMCH.
- ✓ The key advantages of the new neuromuscular model can be summarized as follows:
 - o Presenting a neuromechanical model to start the journey from template to anchors
 - o Presenting a proof of concept to show that considering the body properties (muscle mechanics and control) FMCH could be implemented in human locomotion.
 - o Supporting the stiffness modulation in FMCH, within the neuromuscular control
 - o Introducing LQR as an adaptation method presents a method for higher level control which can increase robustness and efficiency
- ✓ In the revised version, we have clarified that we do not target presenting a more complex model to outperform human-like posture control in the FMCH method, but we explain how the VPP and FMCH concepts may be implemented in biological body.
- ✓ Furthermore, there are advantages achieved by adding muscle model to FMCH that we have shown in the revised version by
 - o Modifying figures 3,4 and 6 by adding FMCH results
 - o Adding new results in Fig. 8 of the revised version showing improved basin of attraction in nmF and adaptive nmF models.
 - o New comparison between the consumed energy in different approaches.
 - o Adding new results about making trade off between efficiency and robustness.

The authors stated that their reflex control is a “bioinspired posture control method” in comparison with Geyer’s (ref 23) use of optimization methods. However, LQR inherently is an optimization based on Q and R . Also, Q and R were chosen as identity. How sensitive are the results when giving more importance to error or effort?

- ✓ Here the output of the LQR controller is not the control effort, but the feedback gain. Therefore, it modulates the muscle force and can be considered in the control effort. Nevertheless, we have added new results to considering different R and Q . In the revised version, the effects of minimizing u and x are compared.

In Section 2.3, please explain what is meant by “fast walking is not reachable” since the reflex-based models tend to have problems with slow speeds. If one speed was chosen for comparison, what was the speed chosen?

- ✓ Due to limitations in SLIP-based model speeds higher than 1.5 m/s are not reachable with the SLIP-based models (without push-off). This is mentioned Sec. 2.3 in the revised version
- ✓ The speed limitation is related to the SLIP model and not the reflex-based control. The model can generate walking from slow (25%PTS) to normal walking speed (75%PTS).
- ✓ Moderate walking speed (1m/s, 50%PTS) was selected as the mean value for the the feasible range. Similar investigations can be done for other reachable speeds.

In comparisons with human data, why is there a time shift in Figure 3? Are model and human walking at the same speed? Toe-off at 80% of gait is quite late. Is there something in the model delaying push-off? Could the authors quantify the difference between model and human trajectories (with cross correlation values perhaps)?

- ✓ Difference between the stance phase duration of the models (78% of gait cycle) and human gait (67% of gait cycle) relates to the current limitations of the model such as having massless and, footless leg.
- ✓ By now the patterns are basically similar to previous SLIP-based model studies. In the recently added Fig. 1 we showed two-segmented leg with biarticular muscles. This could help improving model prediction.
- ✓ One advantage of more complicated models like nmF is resolving what templates cannot explain. However, we cannot resolve all issues at once. Similar patterns at CoM vertical position, GRF and hip torques are addressed in this study.
- ✓ In the future work, we describe the next steps in model extension which could solve this issue.

A few items were brought up in Results with little prior context. For example, more explanation is needed for reflex, such as how was the optimal predefined signal derived? And how was power derived? Is that joint power from hip torque and angular velocity? For Table 3, how was recovery defined (e.g. some number of steps taken after the perturbation)?

- ✓ We have explained the reflex control in more detail in Sec. 2.2.2.
- ✓ AS the only energy management source in our model is at hip torque control, the consumed power and energy are referred to this joint. In the revised version, we have explained calculation of energy consumption with Eq. 25.

- ✓ Recovery is defined based on accomplishing sufficient steps after perturbations and converging to the stable limit cycles (see Fig. 5 and 7). It is described in Sec. 2.2 in the revised version.

On page 16, the authors state the main results from their use of the nmF to model the gait of stroke patients. However, the cited paper does not contain any of the co-authors as authors. Do the authors instead mean that the model can predict what the cited paper found?

- ✓ These studies are not published yet. The results are reported in Balance (Fp7 Eu project) report which is publicly available in the following link: http://www.balance-fp7.eu/private_area/archivo.php?archivo=34.
- ✓ There was a mistake in the previous version and the formulation was confusing. We have corrected it in the current version.

There seems to be three different main points. In the beginning, the authors motivated the use of simpler models to explain complex human behavior. But then they added a complexity to their model by using muscles. Another main point was that understanding and controlling the trunk is vital to gait behavior. Presumably this is one reason why the authors added hip muscles. Then in the discussion and conclusion, the authors discuss the application of the model on exoskeletons by further increasing the complexity, which seems contradictory to the first point.

- ✓ Simplicity is good, but cannot explain all features. We need to increase complexity to address more features. This is the basis of template-anchor concept.
- ✓ Here adding complexity is to show the FMCH model is consistent with the properties of the human body and the neuromuscular system.
- ✓ Without understanding this complex system we cannot assist human locomotion using assistive devices. Therefore, simplified models can help us better understand complexity in biological locomotor system.
- ✓ Finally,, for implementing on robots and exoskeletons, we need to have detailed models.
- ✓ In the revised version, we did our best to clarify the above presented logic by improving explanations and adding a new figure (Fig. 2).

I realize that this is an extension and build up of previous work, but the authors themselves are co-authors of over 1/3 of the references cited (mostly Sharbafi). Perhaps Song and Geyer's 2015 paper on the 3D neuromuscular model should be also included, because his model includes the rectus femoris, reflexes, trunk control, and was tested with push perturbations.

- ✓ We have revised the references, added the suggested reference beside 10 more references (E.g., for reflex control).
- ✓ We removed few of less related self-citations.

Minor comments:

Section 2.2.1- please briefly include a few of the advantages of VBLA over horizontal velocity.

- ✓ We have added the requested description in the revised version. It was already published in a paper presented in Humanoids 2016.

Section 2.2.3 – please define G^ and U^* .*

- ✓ These notations are defined in the revised version of the manuscript.

Section 3 first line – “schems” typo

- ✓ This is corrected in the revised version of the manuscript.

Fig 3: “weigth” typo, add that the dot signifies toe-off, is Figure 3a x-axis label meant to be gait cycle?

- ✓ This is corrected in the revised version of the manuscript.

Appendix B

Positive Leg Force Feedback on Hip Muscles Supports Postural Stability in Bipedal Walking

I appreciate the work the authors have done to address the past comments and am especially satisfied with the additional comparison between nmF and FMCH models. The figures look quite good now and I like the use of a figure to explain templates and anchors. In terms of further suggestions for improvement, I think the abstract and rest of the paper should focus on the main three findings outlined very well in the response to reviewers:

- (i) nmF model reproduce similar mechanics to FMCH model
- (ii) but nmF model is more robust and more efficient
- (iii) so nmF model can be used where FMCH model was being used before

I think the paper can be published with some minor revisions outlined below.

Title

While I appreciate the changes made to the title and think it is more appropriate now, it is still a bit too vague in my opinion. I suggest something a bit more specific. Perhaps along these lines:

“positive leg force feedback on hip muscles during stance predicts stability in simulation for bipedal walking”

Abstract

The abstract is a bit too long, the authors should work to shorten it more.

“Understanding the underlying principles of legged locomotion can be facilitated by developing new conceptual models of human gaits representing the significant features. Such models can be later utilized for design and control of bipedal robots or assistive devices.”

These first two sentences in the abstract are more discussion-oriented than introductory and should be put at the end of the abstract. The first sentence may be removed altogether as it is an extremely general statement about the usefulness of models in gait analysis.

Specify in the abstract that the model is a sagittal plane model

“The simulation results show that both kinematic and dynamic features of the nmF model fairly resemble those observed in human walking.”

This sentence is misleading in that it could be interpreted as nmF model resembles human walking mechanics better than FMCH model. But, in truth, it only does as well as FMCH (as shown in figure 4). Revise to:

“The simulation results show that both kinematic and dynamic features of the nmF model fairly resemble those observed in human walking, similar to the match by the FMCH model.”

Introduction

“The striking feature of templates is that they ignore all the superfluous redundancies”

Please temper the language here. There are many in the field who would fervently disagree with this and believe that the ability of the human body to be redundant is crucial and very far from “superfluous”.

“On the other hand, investigating human/animal locomotion introduces new control concepts such as Virtual Pendulum (VP) [10] which are fundamentally different from engineering techniques”

Unclear statement. Revise or remove.

I don't see the position of current study clearly labeled on the template-anchor scale in figure 1

Methods

The additional explanation of the models are very effective.

Figure 2 has subscript with uppercase H for r_h while text has lowercase h.

Citation on line 167, 174, 175 and 184 were not compiled properly. Shows as ?. Check for this throughout the manuscript

In the methods section, please include what grid size and details of how the basin of attraction was computed. From the figure for the basin of attraction, it seems as though it would benefit from a finer grid.

Results

Justify and provide motivation why the reflex model has been included in this paper in the introduction section. Perhaps include it in the abstract as well? If it is unstable, why is it interesting to compare to?

Divide results section into subheadings and subsections that summarize the major results of the paper.

Discussion

I really like the paragraph that summarizes key findings from lines 579 to 589. I recommend modeling the abstract and the rest of the paper tightly around this in the final edit to maximize impact of the paper.

Conclusion

Perfect. No suggestions.

**Authors' response to the reviewers' comments on the
Paper No. RSOS-181911**

Positive Leg Force Feedback on Hip Muscles Supports Postural Stability in
Bipedal Walking

Dear Dr. *Daley*,

We would like to thank you and the reviewers for the constructive and helpful comments and suggestions. We considered all of them and revised the paper accordingly. To address the comment, suggestions, and concerns, we modified the paper and added new explanations/clarifications.

Editor's comment

Associate Editor Comments to Author (Dr Monica Daley):

Thank you for your patience in waiting for a decision on this manuscript. We have now received feedback from the referees, who are happy with the revised version of the paper, and suggest only a few minor additional changes to the text. I am therefore happy to accept the paper for publication, subject to addressing these final comments. Please ensure that the reference list is correctly updated, as noted by Reviewer 2.

We thank the editors and reviewers for their time and positive feedback. We have revised the paper to address the minor comments. All the changes are shown by blue color in this letter and in the revised manuscript.

The main modifications can be summarized as follows:

- The title of the paper is revised to address the main contribution of this study
- The abstract is revised and shortened.
- Paragraphs with unclear or misleading statements are revised based on the comments
- Missing references are corrected and one reference is added
- Minor changes in Fig. 1 and Fig. 2

Reviewer 1:

I appreciate the work the authors have done to address the past comments and am especially satisfied with the additional comparison between nmF and FMCH models. The figures look quite good now and I like the use of a figure to explain templates and anchors. In terms of further suggestions for improvement, I think the abstract and rest of the paper should focus on the main three findings outlined very well in the response to reviewers:

(i) nmF model reproduce similar mechanics to FMCH model

(ii) but nmF model is more robust and more efficient

(iii) so nmF model can be used where FMCH model was being used before

I think the paper can be published with some minor revisions outlined below.

- ✓ We are thankful for the very precise and solid comments of the reviewer. With respect to this general comment, we made changes in this revision which are explained point by point the following.

Title

While I appreciate the changes made to the title and think it is more appropriate now, it is still a bit too vague in my opinion. I suggest something a bit more specific. Perhaps along these lines:

“positive leg force feedback on hip muscles during stance predicts stability in simulation for bipedal walking”

- ✓ We appreciate the comment from the reviewer. We realized that neither our previous nor the proposed title describe the key insights of the study, as this title could also hold for recent papers on FMCH model.
- ✓ We therefore modified the title in the revised version to be more specific to this work and to describe the main contributions:

“From Template to Anchors: Transfer of VPP balance template to adaptive neuromuscular gait model increases walking stability”

Abstract

The abstract is a bit too long, the authors should work to shorten it more.

- ✓ We have revised the whole abstract based on the suggestion from the reviewer
- ✓ We shortened the abstract in the revised manuscript.

“Understanding the underlying principles of legged locomotion can be facilitated by developing new conceptual models of human gaits representing the significant features. Such models can be later utilized for design and control of bipedal robots or assistive devices.”

These first two sentences in the abstract are more discussion oriented than introductory and should be put at the end of the abstract. The first sentence may be removed altogether as it is an extremely general statement about the usefulness of models in gait analysis.

- ✓ To enter the topic of the paper, we wrote two sentences about modeling to initiate the idea of benefitting from models in design and control. As correctly mentioned by the reviewer, it is too much and for this we replaced these sentences with the following more specific sentence:

“Biomechanical models with different levels of complexity are of advantage to understand the underlying principles of legged locomotion.”

Specify in the abstract that the model is a sagittal plane model

- ✓ Basically, the SLIP model is developed in 2D and there are few studies on 3D in which the models are called 3D-SLIP models. As we mention in the paper that the nmF model is an extension of SLIP by adding trunk and muscles, it means it is in sagittal plane. We could say 2D SLIP model, but then it will be redundant.

“The simulation results show that both kinematic and dynamic features of the nmF model fairly resemble those observed in human walking.”

This sentence is misleading in that it could be interpreted as nmF model resembles human walking mechanics better than FMCH model. But, in truth, it only does as well as FMCH (as shown in figure 4). Revise to:

“The simulation results show that both kinematic and dynamic features of the nmF model fairly resemble those observed in human walking , similar to the match by the FMCH model .”

- ✓ This is corrected in the revised manuscript.

Introduction

Sentence from previous version:

“The striking feature of templates is that they ignore all the superfluous redundancies”

Please temper the language here. There are many in the field who would fervently disagree with this and believe that the ability of the human body to be redundant is crucial and very far from “superfluous”.

- ✓ We agree with the reviewer that the word “superfluous” is a misnomer in here. In fact, what the authors had in mind was that “*templates discard body redundancies that simplifies model which is still able to replicate the basic behaviors*”. Definitely, we do not think that redundancies in human body are useless, as we also use biarticular beside monoarticular actuators in our research.

- ✓ The sentence is revised as follows:

“The striking feature of templates is that they ignore all the redundancies and still lend themselves to be used as simple conceptual models explaining complex problems”

Methods

The additional explanation of the models are very effective.

- ✓ We are happy for addressing the main concerns of the reviewer in our first revision round.

Sentence from previous version:

“On the other hand, investigating human/animal locomotion introduces new control concepts such as Virtual Pendulum (VP) [10] which are fundamentally different from engineering techniques”

Unclear statement. Revise or remove.

- ✓ We have revised this sentence as follows:

“From another perspective, bioinspired control concepts such as Virtual Pendulum (VP) [10] were introduced based on analyzing and modeling human (or animal) gaits.”

I don’t see the position of current study clearly labeled on the template-anchor scale in figure 1

- ✓ In the revised version, we show the current study (D) with a box and explained it in the caption.

The additional explanation of the models are very effective.

- ✓ Thank you!

Figure 2 has subscript with uppercase H for r_h while text has lowercase h.

- ✓ This is corrected in Fig. 2 of the revised manuscript.

Citation on line 167, 174, 175 and 184 were not compiled properly. Shows as ?. Check for this throughout the manuscript.

- ✓ This is corrected in the revised manuscript. We also checked the whole paper for similar errors.

In the methods section, please include what grid size and details of how the basin of attraction was computed. From the figure for the basin of attraction, it seems as though it would benefit from a finer grid.

- ✓ The basin of attraction in this study is computed with a resolution of 1 degree for φ and 0.1 rad/s for $\dot{\varphi}$. But since generating this figure is computationally very expensive (more than 24 hours!), we kept the previous figure in the revised manuscript. Increasing resolution will not change the basic characteristics of the figure.
- ✓ We explained how to compute the basin of attraction in the revised manuscript.

Results

Justify and provide motivation why the reflex model has been included in this paper in the introduction section. Perhaps include it in the abstract as well? If it is unstable, why is it interesting to compare to?

- ✓ The reflex control is not unstable as can be seen in Fig. 5 before perturbation. However, its robustness is much smaller than the reflex based methods. So after perturbations which are not placed in its basin of attraction, it becomes unstable.
- ✓ We need to show that the reflex control which is similar to feedforward approach, is not sufficient and we need feedback signals. Otherwise, the reader might think that no feedback is required at all. The same debate is valid for reflex control and CPG in the literature.
- ✓ One paragraph about reflex control was in Sec. 2.2.2 of the previous manuscript. In the revised version, few sentences are added in the introduction to explain the importance of showing reflex control in this paper.

Divide results section into subheadings and subsections that summarize the major results of the paper.

- ✓ We divided the “Results” section into four subsections as follows:

3.1 Comparison to human walking

3.2 Robustness against external perturbations

3.3 Efficiency and performance in adaptive nmF

3.4 Basin of attraction

Discussion

I really like the paragraph that summarizes key findings from lines 579 to 589. I recommend modeling the abstract and the rest of the paper tightly around this in the final edit to maximize impact of the paper.

- ✓ Thanks for the comment. We revised the abstract, title and parts of the manuscripts based on this line of thought to increase the impact as suggested here.

Conclusion

Perfect. No suggestions.

- ✓ Thank you for your positive feedback.

Reviewer 2:

Comments to the Author(s)

The purpose of this manuscript is to extend a previously developed Force Modulated Compliant Hip (FMCH) "template" model towards a neuromuscular "anchor" model. Whereas the previous model used hip springs to actuate the legs, the new model nmF uses Hill-type muscle models of the rectus femoris and hamstring. Discrete LQR was also used to adapt the reflex parameters during perturbed steps. The model resembles human walking and is an improvement in terms of robustness over the FMCH.

The revised manuscript is a great improvement, and the authors have addressed my major concerns. The motivation of the manuscript is much clearer now (template to anchor models), in the text and with the addition of Figure 1.

- ✓ Thanks for your positive feedback and the nicely summarized description of the work.

A few remaining comments:

Page 10 Line 22 to 41 – This is a confusing paragraph. Since there are two hip muscles in the nMF model, it is unclear how the vastus was included or used. The text also jumps between the current model and other models, which makes it more confusing. Perhaps it would be better to put the details about the authors' model first and then note how it is different from others.

- ✓ We use the leg force (F_s) which is the GRF in the leg axial direction as a feedback signal which is resulted from leg spring force. In the previous version, it is explained that in human body the VAS muscle force can be used to approximate the leg force in our model. This muscle force is measurable and also used in neural control of human gaits. So we do not use VAS force as we do not have segmented leg and we have direct access to F_s in our model.
- ✓ We revised this paragraph for more clarification addressing this comment.

Old references seem to have been submitted in error. There are several citations with question marks. In the response to reviewers, the authors stated they have modified the references, but the references submitted are the same as before.

- ✓ We have corrected all the missing references.

- ✓ In the previous revision, we removed a few of self-citations and added new references. For an unknown reason (probably, a bug in using overleaf for compiling latex) the references were not updated. The changes are listed below:
 - The following references are removed:
 - [9] Sharbafi MA, Maufroy C, Maus HM, Seyfarth A, Ahmadabadi MN, Yazdanpanah MJ. Controllers for robust hopping with upright trunk based on the virtual pendulum concept. In: Intelligent Robots and Systems (IROS), 2012 IEEE/RSJ International Conference on. IEEE; 2012. p. 2222-2227.
 - [13] Sharbafi MA, Ahmadabadi MN, Yazdanpanah MJ, Nejad AM, Seyfarth A. Compliant hip function simplifies control for hopping and running. In: Intelligent Robots and Systems (IROS), 2013 IEEE/RSJ International Conference on. IEEE; 2013. p. 5127-5133.
 - [27] Pratt JE, Krupp BT. Series elastic actuators for legged robots. In: Defense and Security. International Society for Optics and Photonics; 2004. p. 135-144.
 - [33] Oehlke J, Beckerle P, Seyfarth A, Sharbafi MA. Human-like hopping in machines. Biological cybernetics. 2018;p. 1-12.
 - references **no. 17, 18, 19, 20, 23, 27, 31, 32, 37 and 47** were added in the revised manuscript.

Minor comments:

Page 2 Line 50 – It is unclear what is meant by engineering techniques. Can the authors provide examples from the cited works?

- ✓ In the revised manuscript, this sentence is revised as follows:

“From another perspective, bioinspired control concepts such as Virtual Pendulum (VP) [10] were introduced based on analyzing and modeling human (or animal) gaits.”

In Page 5 Line 32 – what does “in the sequel” refer to?

- ✓ We agree with the reviewer that the phrase “in the sequel” seems unnecessary in the text. Therefore, it is omitted in the revised manuscript.

Page 8 Line 14 – “at THE hip joint”

- ✓ “the” is added in the revised version.

Page 9 Line 12 – is 50 steps the authors’ definition or a known standard? It is worded as if it is the standard. Perhaps “... if it can keep stability, specified here as 50 steps after perturbation occurrence” would be clearer. I also suggest using “maintain,” instead of “keep.”

- ✓ Step to fall is a standard method for stability analyses and 25- 50 steps before falling was used in different studies.
- ✓ We revised the paragraph based on this comment.
- ✓ Two references are cited in the modified paragraph.

Page 9 Line 20, Page 10 Line 49 – do not use contractions (“can’t”)

- ✓ These are corrected in the revised manuscript.

Page 11 Line 9 – Perhaps “we compare” is more explicit that “we examine” because the reflex control is later compared against adaptive nmF.

- ✓ This is corrected in the revised manuscript.

Page 12 Line 53 – Note the achievable range, instead of just the upper limit (as the authors have clarified in the response)

- ✓ In the revised version, the achievable range is stated instead of the upper bound.

Page 13 Line 50 – Was stride time the same between model and human?

- ✓ The stride time is shorter than that of human walking at normal speed as the swing phase is shorter because of having massless legs. However, the difference in the duty factor is more important as mentioned in the manuscript.

Page 14 Eqn 25 – I do not think τ_1 and τ_2 were already defined

- ✓ τ_1 and τ_2 are already defined in the text; Page 4, Line 110-112.
- ✓ In the revised version, we have described the terms in Eq. 25, in the last sentence before the equation.

Page 14 Line 49 – Include that both energy and peak power are calculated. Otherwise it is confusing why the equation for work is shown, but the results are in terms of power.

✓ This is added in the revised manuscript.

Page 1 – Merely to further assist the reader: adding labels (BSLIP, BTSLIP, FMCH, nMF) onto the A-D figures.

✓ The labels are added in the revised manuscript.